# Oral immune dysfunction is associated with the expansion of FOXP3+PD-1+Amphiregulin+ T cells during HIV infection

N. Bhaskaran[1], E. Schneider[1], F. Faddoul[2], A. Paes da Silva[3], R. Asaad[4], A. Talla[5], N. Greenspan[5], A. D. Levine [6], D. McDonald[7], J. Karn [6,8], M. M. Lederman[4,5] & P. Pandiyan [1,5,8 ✉]

Residual systemic inflammation and mucosal immune dysfunction persist in people living with HIV, despite treatment with combined anti-retroviral therapy, but the underlying immune mechanisms are poorly understood. Here we report that the altered immune landscape of the oral mucosa of HIV-positive patients on therapy involves increased TLR and inflammasome signaling, localized CD4+ T cell hyperactivation, and, counterintuitively, enrichment of FOXP3+ T cells. HIV infection of oral tonsil cultures in vitro causes an increase in FOXP3+ T cells expressing PD-1, IFN-γ, Amphiregulin and IL-10. These cells persist even in the presence of anti-retroviral drugs, and further expand when stimulated by TLR2 ligands and IL-1β. Mechanistically, IL-1β upregulates PD-1 expression via AKT signaling, and PD-1 stabilizes FOXP3 and Amphiregulin through a mechanism involving asparaginyl endopeptidase, resulting in FOXP3+ cells that are incapable of suppressing CD4+ T cells in vitro. The FOXP3+ T cells that are abundant in HIV-positive patients are phenotypically similar to the in vitro cultured, HIV-responsive FOXP3+ T cells, and their presence strongly correlates with CD4+ T cell hyper-activation. This suggests that FOXP3+ T cell dysregulation might play a role in the mucosal immune dysfunction of HIV patients on therapy.

[1] Department of Biological Sciences, School of Dental Medicine, Case Western Reserve University, Cleveland, OH, USA. [2] Advanced Education in General Dentistry, School of Dental Medicine, Case Western Reserve University, Cleveland, OH, USA. [3] Department of Periodontics, School of Dental Medicine, Case Western Reserve University, Cleveland, OH, USA. [4] University Hospitals Cleveland Medical Center AIDS Clinical Trials Unit, Division of Infectious Diseases & HIV Medicine, Cleveland, OH, USA. [5] Department of Pathology, School of Medicine, Case Western Reserve University, Cleveland, OH, USA. [6] Department of Microbiology and Molecular Biology, School of Medicine, Case Western Reserve University, Cleveland, OH, USA. [7] Division of AIDS, NIAID, NIH, Bethesda, MD, USA. [8] Center for AIDS Research, School of Medicine, Case Western Reserve University, Cleveland, OH, USA. ✉email: pxp226@case.edu

Human immunodeficiency virus 1 (HIV-1)-associated co-morbidities such as inflammatory disorders and cancer are important public health concerns[1–6]. Immune complications persist in patients despite effective combined anti-retroviral therapy (cART) and have been inextricably linked to HIV latency, altered mucosal T cell functionality, and increased production of immune activation-associated cytokines in treated people living with HIV (PLWH)[7–14]. Although oral complications such as periodontitis and oropharyngeal cancer in healthy HIV-uninfected adults are usually mild, self-limited, and of short duration, they are of increased incidence and severity in HIV+ individuals under suppressive HIV therapy[15–18]. Oral mucosa is conferred with a distinct immune compartment with a unique microbiome[19], but the oral lymphoid cell population and its dysregulation in HIV+ patients are not understood[18,20,21]. Acute simian immunodeficiency virus (SIV) infection has been shown to cause a loss of barrier protection as a result of CD4+ T cell depletion in oral mucosa[22]. Although cART therapy can restore these CD4+ T cells, they can contribute to oral viral reservoirs. To date, there is no information on alterations of oral mucosal CD4+ T cell functionality in the context of SIV or HIV infection after treatment. CD4+CD25+Foxp3+ regulatory T (Treg) cells, known for their immunomodulatory functions, express C-X-C chemokine motif receptor 4 (CXCR4) and C-C chemokine receptor 5 (CCR5) coreceptors and support high levels of HIV infection and replication[23]. Thus, the initial loss of Tregs during HIV infection can contribute to a self-perpetuating loop of events leading to immune activation[7,24–28]. Previous reports document varied levels of Tregs, depending on the location (blood, lymphoid organ, or mucosa) and acute versus chronic phase of infection[29–34]. Nevertheless, the precise cellular and functional alterations in Tregs in the context of immune activation have not been characterized in the oral mucosa.

Here we show that gingival mucosa of treated PLWH has an increased accrual of PD-1hiIFN-γ+FOXP3+ cells with an elevated expression of amphiregulin (AREG), BCL-2, and interleukin (IL)-10 when compared to healthy individuals. Counterintuitively, it also reveals upregulation of Toll-like receptor (TLR) and inflammasome pathways, with CD4+ T cells showing hyperactivated phenotype. Mechanistically, the FOXP3+ cells require IL-1β and programmed cell death protein 1 (PD-1)-dependent asparaginyl endopeptidase (AEP) activation for expansion and sustained expression of FOXP3 and AREG. However, these cells lack suppressive function in vitro, implying an impairment of Treg-mediated mucosal CD4+ T cell homeostasis in HIV+ patients.

## Results

**Oral gingival tissue displays inflammatory signature and CD4 T cell alterations in HIV+ patients on therapy.** To examine immune cell alterations in oral mucosa, we recruited 78 participants that included healthy controls and treated HIV+ (HIV+ cART) individuals and collected their saliva, peripheral blood mononuclear cells (PBMCs), and oral gingival mucosa biopsies (Supplementary Table 1). Unbiased RNA sequencing (RNA-seq) analyses revealed an upregulation of 772 transcripts and down-regulation of 226 transcripts in gingival biopsy tissues of HIV+ cART individuals when compared to controls (Fig. 1A, left). However, only 54 genes were differentially regulated in their PBMCs (Fig. 1A, right), indicating a mucosal dysfunction persisting during therapy after significant clearance of the virus. Global pathway analysis identified that a majority of the upregulated genes in oral mucosa of HIV+ patients were associated with TLR, myeloid differentiation factor 88 (MyD88), inflammasome, and inflammatory responses, highlighting an underlying

immune activation (Fig. 1B–D). Gene set enrichment analysis (GSEA)[35] revealed a positive enrichment of pathways of aging, head and neck cancer, and AKT1 signaling based on gene sets in gene ontology (GO) pathways and MSigDB (Supplementary Fig. 1A). The frequency of CD38 and HLA-DR co-expressing cells, the hallmark of HIV-mediated CD4+ T cell activation[36], was significantly higher in human oral intraepithelial and lamina propria leukocytes (HOILs) from gingival biopsies from HIV+ patients on therapy (Fig. 1E and Supplementary Fig. 1B, C). As we have shown previously, there were no differences in the frequency of activated CD4+ T cells between the PBMCs of the groups[36]. Neither were there any differences in overall CD4+ T cell proportions or the levels of interferon (IFN)-γ-expressing CD4+ cells between these groups (Supplementary Fig. 2A–C). Collectively, peripheral CD4+ T cells appear to be largely restored by cART, but the oral mucosa of HIV+ patients display features of immune dysregulation with alterations in inflammasome pathway, TLR/MyD88 signaling, and localized CD4+ T lymphocyte hyperactivation.

**CD4+CD25+FOXP3+ cells are enriched in oral mucosa of HIV+ patients on therapy.** Based on the upregulation of transcripts in the inflammasome pathway (Fig. 1D), we then assessed the IL-1β levels in HIV+ patients. We have previously shown increased IL-1β in lymphoid organs of HIV+ patients[37], but the oral mucosa has not been examined. While IL-1β levels appeared to be lower (Supplementary Fig. 3A), IL-6 levels were significantly higher in the saliva of HIV+ cART patients (Supplementary Fig. 3B). We also determined their expression in the supernatants of stimulated oral gingival immune cells ex vivo. These cells derived from HIV+ patients showed significantly elevated levels of secreted IL-1β and IL-6, corroborating with their inflammatory signature (Figs. 1C, D and 2A). Given the role of microbial ligands in regulating mucosal cytokines, we hypothesized that dysbiotic oral microbiome may also be linked to alterations in cytokine levels and TLR signaling in oral mucosa of HIV+ patients[38,39]. We found that salivary soluble TLR2 proteins were significantly increased in HIV+ patients (Fig. 2B). Interestingly, younger HIV+ patients (<60 years) showed increased levels of sCD14 in their serum compared to young healthy controls (Supplementary Fig. 3C). These features of inflammation, i.e., CD4 hyperactivation (CD38+ HLA-DR+) and alterations in TLR2 signaling led us to hypothesize that there might be a defect in immune regulation in oral mucosa of HIV+ patients[40,41]. By first examining the transcriptome of gingival mucosa for the genes involved in promoting Treg development and functions, we found that some of the Treg transcripts were significantly enriched in oral mucosa of HIV+ patients (Fig. 2C). Flow cytometric analyses of CD4, CD25, and FOXP3 expression also revealed that oral mucosal Treg proportions were strikingly higher in the HIV+ group compared to the HIV-negative individuals (Fig. 2D, E, top). However, there were no differences in Treg percentages in PBMCs between these two groups, showing that Treg dysregulation was specific to the mucosa (Fig. 2D, E, bottom). Because CD4+ T cells exhibited a hyperactivated phenotype (Fig. 1E), we anticipated a lower frequency of FOXP3+ T cells in the HIV+ group but were surprised to find increased Treg proportions in oral mucosa of HIV+ individuals. Increased TLR2 signaling that we observed in HIV+ patients (Figs. 1B, D and 2B) can enhance FOXP3+ cell proliferation and alter the functions of Treg and non-Treg CD4+ T cells[42,43]. It is known that oral complications such as periodontitis are of increased incidence and severity in HIV+ individuals even after suppressive HIV therapy. A majority of the HIV+ patients in our cohort had previous oral lesions. Therefore, it is possible that generalized inflammation such as periodontitis contributes to Treg dysregulation. To verify this possibility, we profiled FOXP3+ cells in gingival

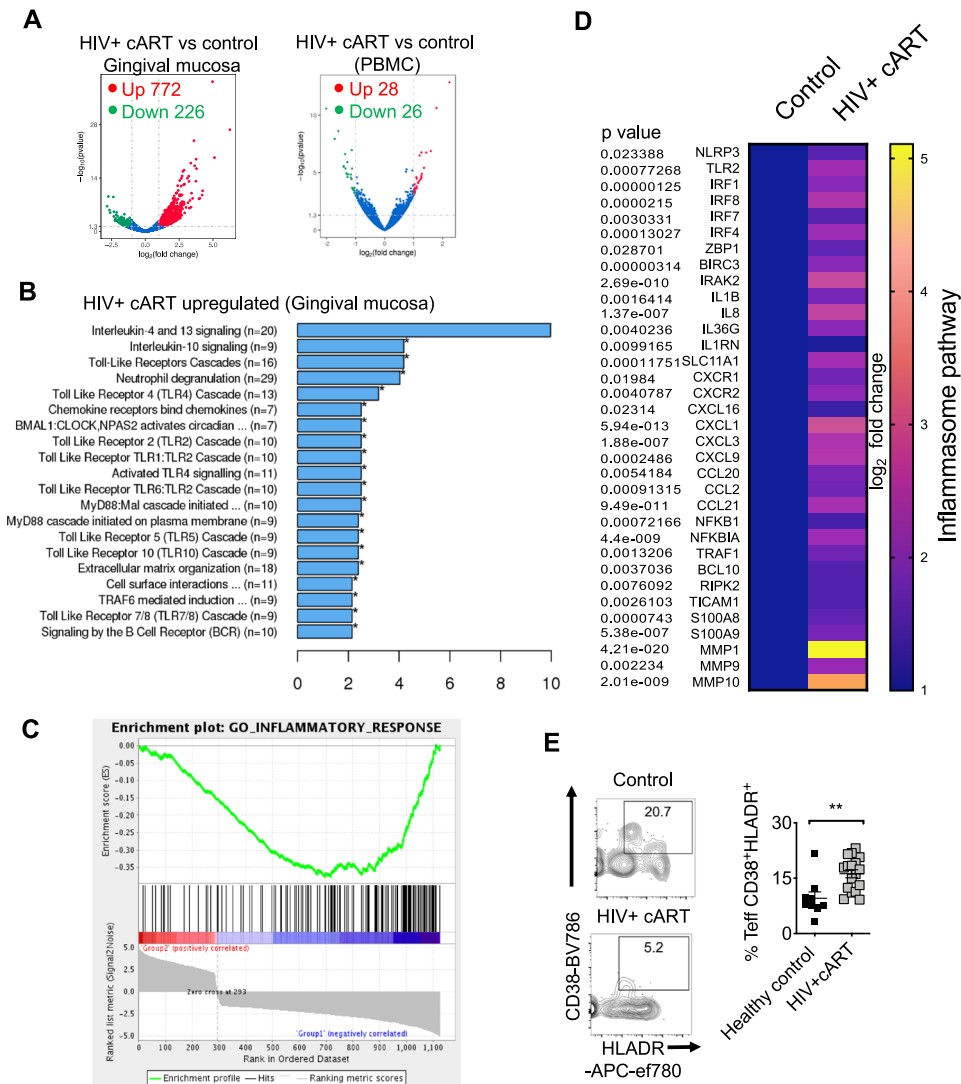

**Fig. 1 Transcriptomic profiling and flow cytometric analysis of oral mucosa in HIV+ patients.** Forty-six HIV+ patients on cART treatment and 32 uninfected healthy controls were recruited (Supplementary Table 1). RNA sequencing was performed in gingival tissues and PBMCs collected from six randomly chosen age-matched participants; healthy uninfected control ($n = 3$) and HIV+ cART ($n = 3$), 2 males and 1 female in each group. Gingival cells were enriched for immune cells by reducing the epithelial cells through gradient centrifugation before transcriptome analyses. Volcano plots showing differential RNA expression in HIV+ cART versus healthy uninfected control groups in gingival mucosa (**A**, left) and PBMCs (**A**, right). **B** REACTOME pathway analysis of the genes upregulated in HIV+ cART gingival mucosa. **C** Gene set enrich analysis (GSEA) was performed using the GSEA software (Broad Institute; http://www.broad.mit.edu/GSEA) employing the entire gene list generated from transcriptome analyses. This whole gene list was pre-ranked based on T Score and then uploaded to GSEA software. Inflammatory response signature genes were defined based on the gene sets in MSigDB. **D** Heatmaps showing upregulation of inflammasome signature genes that were defined based on the published literature. Human oral intraepithelial and lamina propria leukocytes (HOILs) from gingival biopsies were processed for flow cytometry. **E** Effector CD4 cells were gated as shown in Supplementary Fig. 1B and further on FOXP3-negative population. Contour plots (left) and statistics (right) showing the percentage of activated (CD38+ and HLADR+) effector CD4+ cells ($n = 20$); mean value ± SEM are plotted. (**P = 0.0029; two-tailed; Mann–Whitney test). Source data are provided as a Source data file.

mucosa from both chronic and acute periodontitis non-HIV patients comparing them with healthy individuals. Although we found increases in T helper type 17 (Th17) cells in periodontitis non-HIV patients as shown previously[44], there were no significant changes in the frequency of FOXP3+ cells in their gingiva (Supplementary Fig. 4). These results show that previous inflammation does not by itself correlate or contribute to FOXP3+ T cell enrichment in the oral mucosa. Taken together, these data raise the possibility that enrichment of FOXP3+ cells might be linked to the upregulation of inflammasome and TLR/MyD88 signaling (Fig. 1B, D) and localized CD4+ T hyperactivation (Fig. 1E) specific to the oral mucosa of HIV+ patients.

**HIV infection of oral mucosa-associated lymphoid tissue (MALT) induces cell death and phenotypic changes in FOXP3+ cells.** The oral mucosal system is composed of compartmentalized MALT, which includes palatine tonsils. The lymphoid environment of the tonsil oral MALT makes these tissues highly susceptible to infection and establishment of HIV reservoirs[45,46]; however, CD4+ T cell dysfunction in relation to oral residual immune activation in cART-treated patients has not been studied before. To obtain mechanistic details underlying $T_{reg}$ alterations during HIV infection, we employed human tonsil cultures (HTC) derived from uninfected individuals. We hypothesized that this system would provide mechanistic insights into immune dysfunction in oral mucosa of HIV

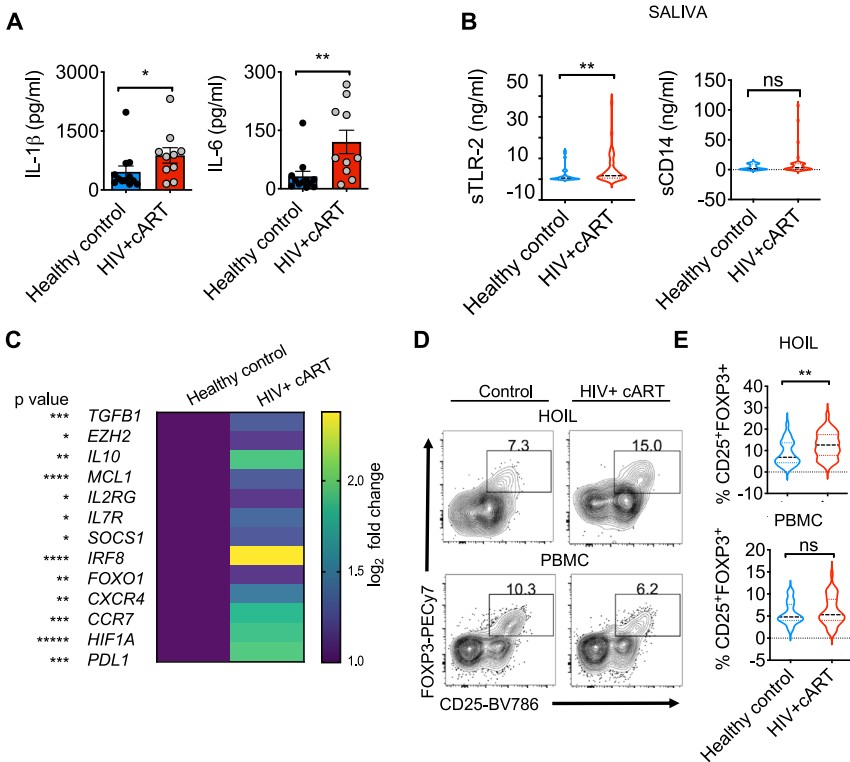

**Fig. 2 Inflammatory cytokines, s-TLR2, and CD4⁺CD25⁺FOXP3⁺ cells are enriched in gingival mucosa of HIV⁺ patients on therapy. A** Cells from gingival mucosa were re-stimulated with PMA/Ionomycin for 4 h and supernatants were collected for ELISA analyses of IL-1β (left) and IL-6 (right) (Control, n = 13; HIV+ cART, n = 10; *P = 0.02; one-tailed; Mann–Whitney test). **B** ELISA quantification of s-TLR2 (left) and s-CD14 (right) levels in saliva (Control, n = 32; HIV+ cART, n = 46; **P = 0.007; two-tailed; Mann–Whitney test). **C** Transcriptome profiling was performed as in Fig. 1. Heatmaps of genes encoding literature-curated T_reg signature proteins differentially regulated in gingival mucosa. Flow cytometric analyses of CD45⁺CD3⁺CD4⁺ gated HOIL cells for CD25⁺FOXP3⁺ cell proportions, showing representative contour plots (**D**), and statistical analysis of T_reg proportions (**E**) in HOILs (above) and PBMCs (below) (Control, n = 31; HIV+ cART, n = 44). Mean value ± SEM are plotted. (**P = 0.0094; two-tailed; Mann–Whitney test). Source data are provided as a Source data file.

⁺ patients in vivo. First, we performed immunophenotyping of the tonsils that were obtained from tonsillectomy surgeries in children. As expected, examining the disaggregated tonsil cells in comparison with PBMCs from independent healthy donors ex vivo, we found that tonsils had comparable levels of CD4⁺ T cells and reduced CD8⁺ T cells (Supplementary Fig. 5A, B, D). They harbored modestly lower frequencies of CD3⁺ T cells, natural killer (NK) and NKT cells, but significantly higher proportions of CD19⁺ B cells (Supplementary Fig. 5B, D). As expected, 25–55% of CD4⁺ T cells were CXCR5⁺PD-1^high follicular T helper cells (TFH) in tonsils (Supplementary Fig. 5C, top, and Supplementary Fig. 5E). The overall proportions of CD4⁺FOXP3⁺ cells were comparable to those in PBMCs, but there were slightly lower proportions of CD25⁺FOXP3⁺ T follicular regulatory cells (TFR) and significantly higher proportions of CD45RO⁺ memory cells in tonsils when compared to PBMCs (Supplementary Fig. 5C, middle, and bottom, and Supplementary Fig. 5E). We stimulated whole tonsil cultures and infected them with the X4- tropic HIV-1 strain NLGNef[47,48]. We characterized the productively infected cells by examining the green fluorescent protein (GFP)-expressing cells (Supplementary Fig. 6A). As shown before[46], 95% of the productively infected cells were of germinal center TFH phenotype (Supplementary Fig. 6B, top) but co-expressed FOXP3 and CD45RO (Supplementary Fig. 6B, middle). Though T_regs have been previously shown to be an HIV-permissible population[49], our results show that these p24 expressing cells also co-express CD45RO (Supplementary Fig. 6B, middle, bottom). They also expressed higher levels of CXCR4, with many co-expressing CCR5, consistent with a previous report on tonsillar CD4⁺ T cells[50].

Examining the FOXP3⁺ cell fraction more closely, PD-1^high-CXCR5⁺CD45RO⁺ cells, which were consistent with the memory phenotype germinal center follicular regulatory cells (TFR), displayed a significant high permissibility to HIV infection (Supplementary Fig. 6C, D). As shown previously[45], these TFR showed higher permissibility to both X4 and R5 tropic viruses. SIV has been previously shown to upregulate the expression of surface and intracellular IL-10R in jejunum lamina propria but not in jejunum intraepithelial T cells during acute SIV infection[51]. Therefore, we examined the expression of the IL-10R receptor but did not find changes in IL-10R expression with and without anti-retroviral inhibitor in HIV-infected CD4⁺ T cells (Supplementary Fig. 6E). To distinguish whether these permissible FOXP3⁺ cells were pre-existing tonsil T_regs or induced during TCR stimulation, we purified CD4⁺ cells and CD4⁺CD25⁺CD127^low cells and infected them with HIV. We found that FOXP3⁺ cells were highly permissible in both of the cultures, although purified T_regs harbored a significantly higher frequency of GFP⁺ cells (Fig. 3A). Similar results were obtained even in the absence of TCR activation, as tonsil cells do not require exogenous stimulation prior to infection[45]. These results confirm previous studies showing high permissibility of FOXP3⁺ cells to HIV infection[45,49]. Moreover, we found that the proportions of FOXP3⁺, as well as FOXP3^negative IL-17A⁺ and IFN-γ⁺ effector CD4⁺ populations, are decreased in HIV-infected tonsils (Fig. 3B and Supplementary Fig. 7A, B), showing that CD4⁺ cells are also highly susceptible to cell death during acute HIV infection (Fig. 3C). This is consistent with previous results on HIV-mediated apoptotic and pyroptotic CD4⁺ T cell depletion[50,52]. Interestingly, we found that

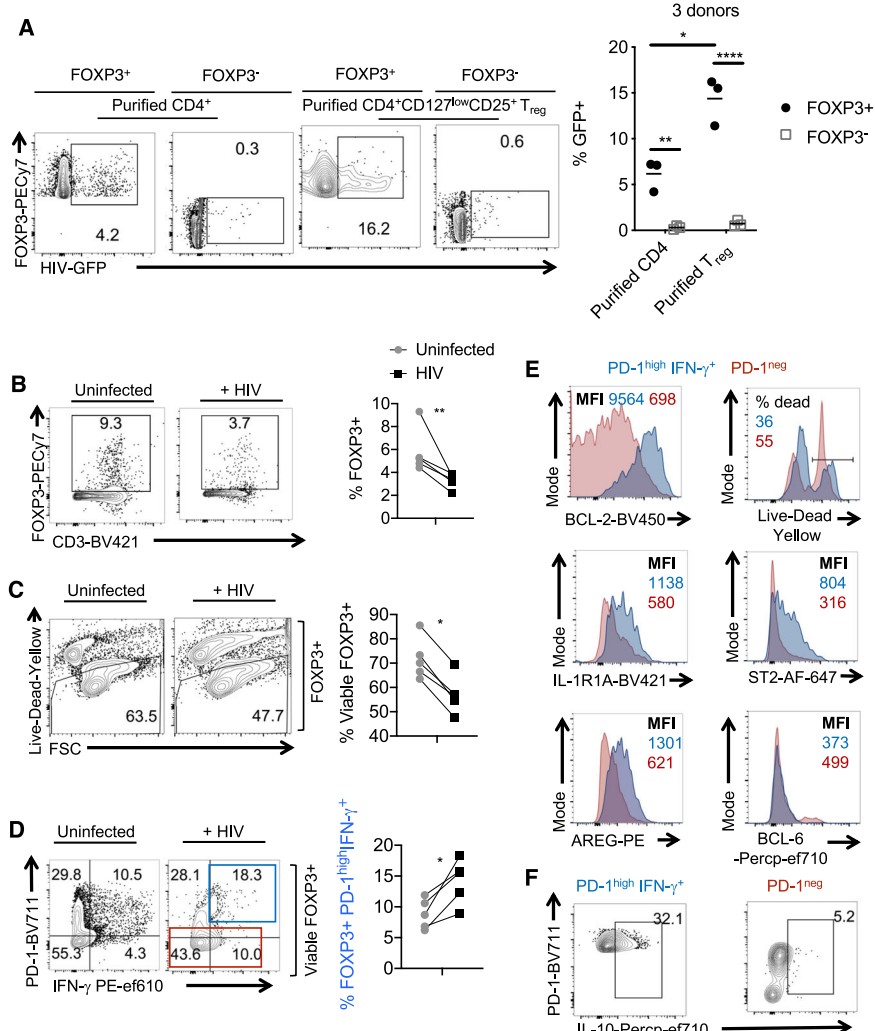

**Fig. 3 HIV infection reduces FOXP3+ cells but increases the proportions of PD-1^high IFN-γ+ cells among FOXP3+ in vitro. A** Purified tonsil CD4+ T cells (~91% purity) or CD4+CD25+ CD127^low T_reg cells (>88% FOXP3+) were TCR activated and infected with HIV as described in "Methods." GFP was assessed in FOXP3+ (left) or FOXP3− (right; gated on Foxp3^neg cells) fractions 48 h post-infection. Representative flow cytometric data (left) and statistical analyses from three independent tonsil donors (right) are shown. Mean +/− SEM are shown; *P < 0.05; **<0.005, ****<0.00005; two-tailed; Mann–Whitney test. **B–F** TCR activated whole human tonsil cultures (HTC) were infected with HIV and allowed to expand with IL-2 for 6 days. Flow cytometric analyses of CD3+FOXP3+ cells pre-gated on CD8− cells (**B**), viability of CD3+CD8−FOXP3+ cells (**C**), PD-1 and IFN-γ expression in viable CD3+FOXP3+CD8− cells (**D**), with respective statistical analyses (right) are shown. **B–D** n = 5; *P < 0.05; **P = 0.007; two-tailed; Mann–Whitney test. **E**, **F** Flow cytometric plots showing the expression of the indicated proteins in PD-1^high and PD-1^low populations gated in **D** in HIV-infected HTC. **B–F** Five independent experiments showed similar results.

the frequency of PD-1^hi IFN-γ+ cells among the viable FOXP3+ population consistently increased with HIV+ infection (Fig. 3D and Supplementary Fig. 8; staining controls). Although this population was productively infected (Supplementary Fig. 9A), it expressed high levels of BCL-2 and was more resistant to cell death compared to PD-1^low cells (Fig. 3E, top). Further characterization of this population revealed that they expressed high levels of CD25 (Supplementary Fig. 9B), IL-1 family receptors such as IL-1R, IL-33R (Suppression of Tumorigenicity 2 (ST-2)), and AREG (Fig. 3E, middle and bottom), resembling activated tissue T_regs. While this population had slightly lower levels of BCL-6, they expressed IL-10 (Fig. 3E, bottom, and Fig. 3F) and B lymphocyte-induced maturation protein 1 (BLIMP1), characteristic of effector TFR cells in germinal centers and tissue T_regs (Supplementary Fig. 9C)[53]. Taken together, these data revealed that, although HIV infection led to the loss of CD4+ T cells, it resulted in an increase of a unique population of PD-1^hi IFN-γ+ AREG+ FOXP3+ cells that survived the infection and might contribute to immune dysfunction.

**Blocking subsequent rounds of infection and cell death increased the proliferation of PD-1^hi IFN-γ+ FOXP3+ cells**. We further characterized the conditions under which these FOXP3+ cells were induced during HIV infection and tested whether mechanisms underlying HIV-induced cell death might play a role. Therefore, we aimed to block cell death by inhibiting HIV replication and caspase activation after the onset of initial cell death. To this end, we added reverse transcriptase inhibitor Efavirenz and pan-caspase inhibitor 28 h after HIV infection. We found that both were able to increase the overall viability of CD4+ T cells including FOXP3+ cells (Fig. 4A, B). Interestingly, while the proportions of PD-1^hi IFN-γ+ FOXP3+ cells were partially reduced by these inhibitors, their absolute cell numbers significantly increased in the cultures (Fig. 4C). These data show that PD-1^hi IFN-γ+ FOXP3+ cells that were induced during initial HIV infection had a survival advantage and likely expanded in the presence of these inhibitors in oral MALT. Consistent with this notion, while PD-1^low FOXP3+ cells did not proliferate much, and the percentage and absolute

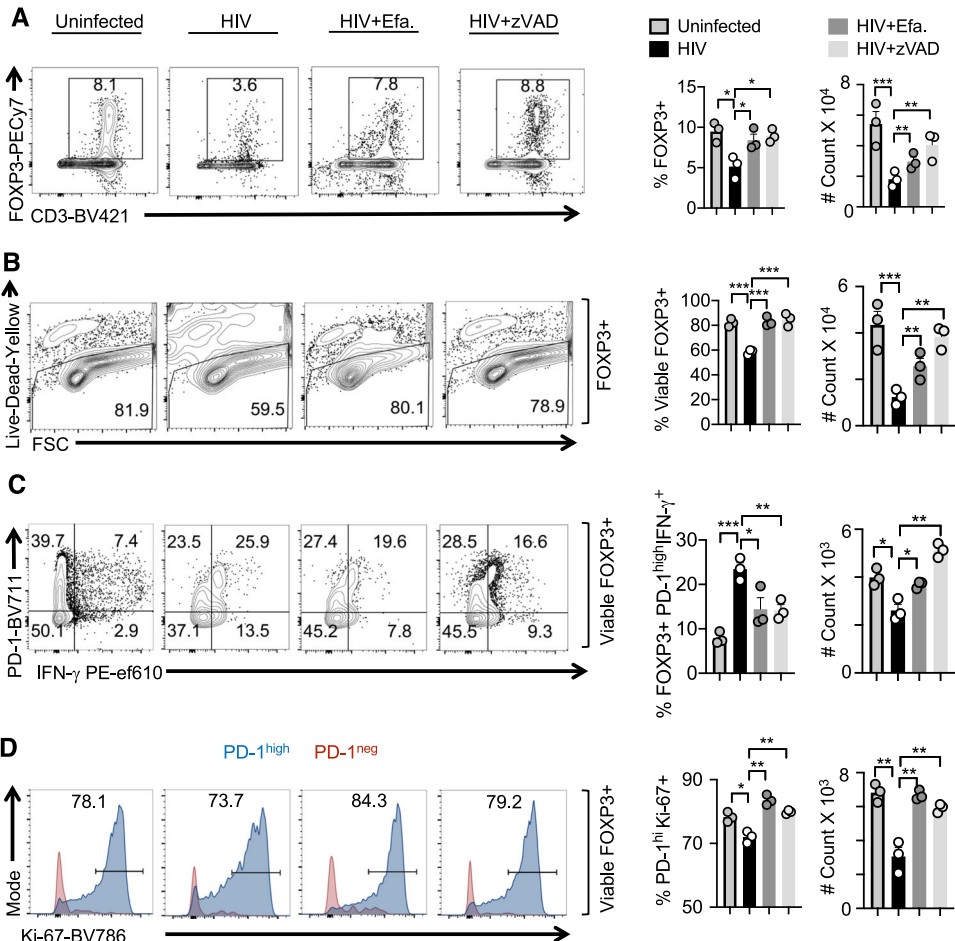

**Fig. 4 Blocking subsequent rounds of infection and cell death increased the proliferation of PD-1hiIFN-γ+FOXP3+ cells.** Whole HTC was activated with TCR stimulation, infected with HIV, and allowed to expand in the presence of TGF-β1 (10 ng/ml) and IL-2 (100 U/ml) for 6 days. Viral inhibitor Efavirenz (50 nM) or cell death/pan-caspase inhibitor z-VAD (10 µM) was added 28 h post-infection as described in "Methods." Flow cytometry acquisition was done with constant time for all the samples. Percentage of CD3+FOXP3+ cells pre-gated on CD8− cells (**A**), viability of FOXP3+ cells pre-gated on CD3+CD8− cells (**B**), PD-1 and IFN-γ expression in viable CD3+CD8−FOXP3+ cells (**C**), Ki-67 expression in viable PD-1high and PD-1low FOXP3+ populations (**D**) are shown. **A–D** Representative contour plots (left), statistical analyses of proportions of the cells (middle), and statistical analyses of the absolute cell counts (right) are shown (ordinary one-way ANOVA alpha = 0.05* and multiple comparison t tests; *$P < 0.05$; **<0.005, ***<0.0005). Results are derived from three independent experiments ($n = 3$) and are presented as mean value +/− SEM.

numbers of Ki-67+ cells were higher in PD-1hiFOXP3+ cells in the presence of these inhibitors (Fig. 4D). Collectively, these data highlight that initial HIV infection is sufficient for PD-1hiIFN-γ+FOXP3+ cell accumulation, and these cells are not abolished with the antiviral drug treatment. Instead, blocking HIV replication and HIV-induced cell death after the initial rounds of HIV infection promoted the proliferation of PD-1hiIFN-γ+FOXP3+ cells that were rescued from cell death.

**PD-1hiIFN-γ+FOXP3+ cell accumulation is associated with the expression of IL-1β-dependent AKT1 signaling and enhanced by TLR2 ligands in the context of HIV infection.** We and others have previously shown that direct and indirect TLR2 signaling in FOXP3+ cells can induce proliferation impacting their functions[40]. Moreover, the results from HIV+ patients that showed TLR2 pathway upregulation and cytokine inflammatory pathways in the oral mucosa (Figs. 1 and 2) led us to hypothesize that TLR2 signaling is involved in PD-1hiIFN-γ+FOXP3+ cell induction. There is copious evidence that HIV+ patients have episodes of recurring oral *Candida* infections and periodontitis despite therapy (Supplementary Table 1), which might contribute

to the enrichment of transcripts involved in TLR signaling in their oral mucosa (Figs. 1 and 2)[15,54,55]. To this end, we determined the effect of lipopolysaccharide (LPS) and TLR2 ligands such as *Candida* (heat-killed germ tube (HKGT)) and *Porphyromonas gingivalis* (PG-LPS) on purified tonsil CD4+ cells in the context of HIV infection. While HKGT moderately increased PD-1hiIFN-γ+FOXP3+ cells, these ligands did not alter cell viability or expansion of PD-1hiIFN-γ+FOXP3+ cells in uninfected cultures (Fig. 5A). However, in HIV-infected cultures, these ligands promoted a significant increase in PD-1hiIFN-γ+FOXP3+ cells, as well as AREG expression in FOXP3+ cells (Fig. 5A, left and right, and Supplementary Fig. 10). We saw consistent results even in the absence of TCR stimulation of CD4+ T cells (Supplementary Fig. 11A, B). To determine the mechanism underlying the accumulation of PD-1hiIFN-γ+FOXP3+ cells and AREG expression in these cells, we examined the cytokine production in cultures. A previous study has shown that HIV induces the secretion of pyroptosis-related cytokine IL-1β in CD4+ T cells[50]. Based on the upregulation of IL-1β and IL-6 in oral mucosa of HIV+ patients (Fig. 2A) and the role of IL-1 family cytokines in promoting AREG expression[56,57], we examined the effect of IL-1β, IL-33, and IL-6 in HIV-infected tonsil CD4+ T cell cultures.

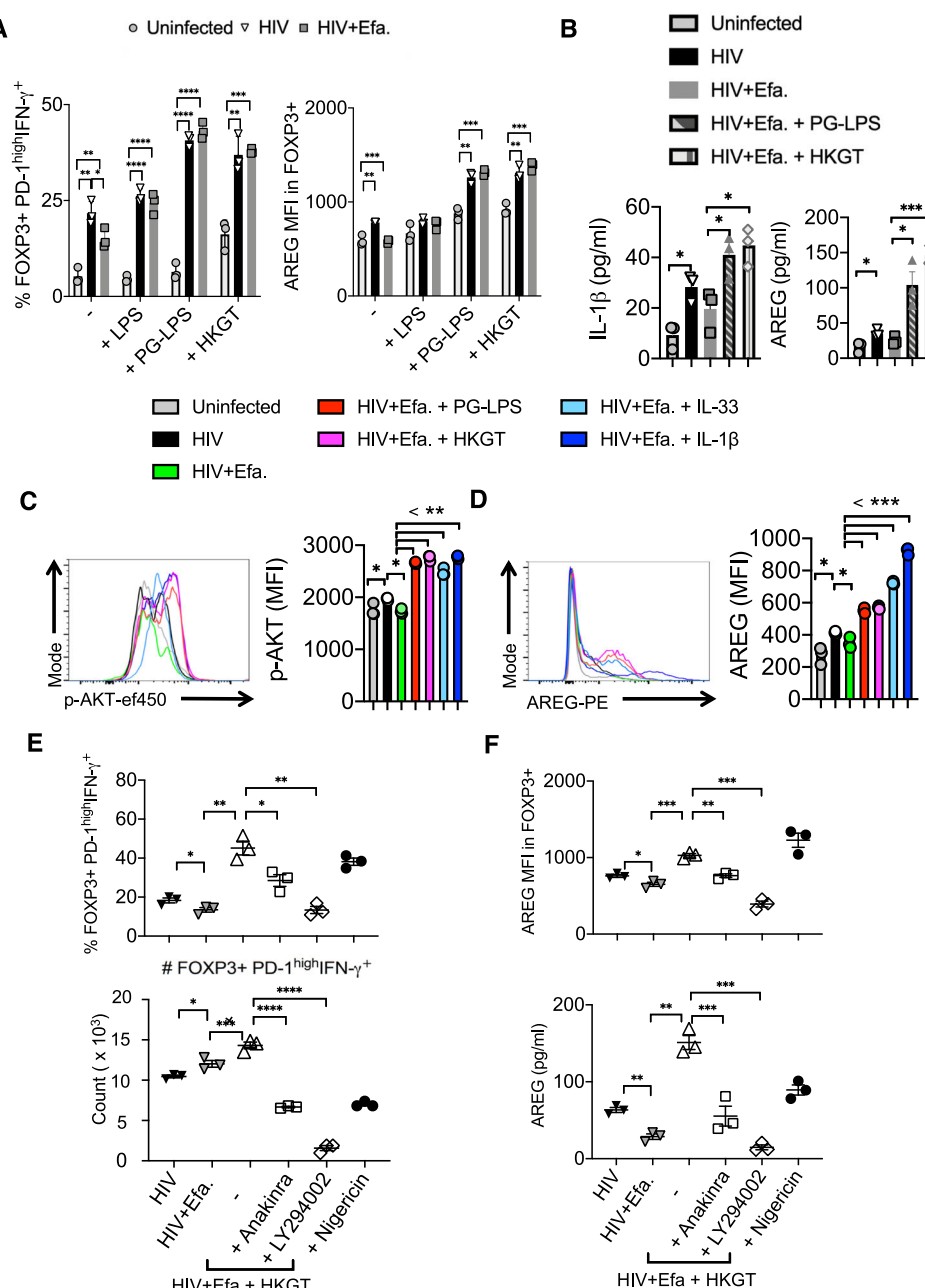

**Fig. 5 PD-1$^{hi}$IFN-$\gamma^+$ FOXP3$^+$ cell induction is associated with the expression of IL-1$\beta$-dependent AKT1 signaling and enhanced by TLR2 ligands in the context of HIV infection.** Purified tonsil CD4$^+$ T cells (~93% purity) were TCR activated, infected with HIV, and allowed to expand in the presence of TGF-$\beta$1 (10 ng/ml) and IL-2 (100 U/ml) for 6 days post-infection unless otherwise noted. Efavirenz (50 nM), LPS (10 μg/ml), PG-LPS (5 μg/ml), HKGT (10$^6$/ml), IL-1$\beta$ (20 ng/ml), IL-33 (20 ng/ml), Anakinra (10 μg/ml), LY294002 (10 μM), and Nigericin (10 nM) were added as indicated, 36 h post infection. **A** PD-1 and IFN-$\gamma$ (left) and AREG (right) expression in FOXP3$^+$ cells. **B** ELISA quantification of IL-1$\beta$ (left) and AREG (right) in cell culture supernatants collected on day 3 post infection. p-Akt (**C**) and AREG (**D**) expression in FOXP3$^+$ cells. **E** Percentage and absolute cell numbers of PD-1$^{hi}$IFN-$\gamma^+$FOXP3$^+$ cells in CD4$^+$ population. **F** AREG expression in FOXP3$^+$ cells (left) and ELISA quantification of AREG (right), 6 days post infection. **A–F** Data are representative of three independent experiments and are presented as mean value $+/-$ SEM. (*$P < 0.05$; **<0.005, ***<0.0005, ****<0.00005; unpaired $t$ tests). Source data are provided as a Source data file.

Enzyme-linked immunosorbent assay (ELISA) quantification demonstrated that HIV infection elevated the levels of mature IL-1$\beta$ and AREG, which were further increased when CD4$^+$ T cells were stimulated with TLR2 ligands (Fig. 5B). While HIV infection did not alter IL-33 and IL-6, it enhanced IL-1$\beta$, which was further upregulated by TLR2 ligands (Supplementary Fig. 11C, D). Induction of mature IL-1$\beta$ is likely caspase-1 dependent, and this cytokine can function in CD4 intrinsic and phosphatidylinositol-3-OH kinase (PI-3K)/AKT1-dependent manner in effector CD4$^+$ T cells[50,58–61]. Moreover, AKT-1 activation/phosphorylation and FOXO3 repression promote activated T$_{reg}$ cell accumulation in tissues[62]. Indeed, we found that HIV infection was able to activate caspase-1, as measured by its phosphorylation in FOXP3$^+$ cells (Supplementary Fig. 12). TLR2 ligands further enhanced caspase-1 activation almost to the levels of Nigericin, an IL-1/inflammasome, and pyroptosis activator (Supplementary Fig. 12).

Next, we examined whether TLR2 ligands or IL-1β can promote PD-1$^{hi}$IFN-γ$^+$FOXP3$^+$ cells and AREG expression induction from naive T$_{regs}$ in the context of HIV infection. About 82–92% of FOXP3$^+$ cells in tonsils are of CD45RO$^+$CD62L$^{low}$ phenotype (Supplementary Fig. 13A). As expected, CD45RO$^{neg}$ naive T$_{regs}$ were of CD62L$^{high}$ and PD-1$^{neg}$ phenotype (Supplementary Fig. 13A). To address whether PD-1$^{high}$ T$_{regs}$ can be induced from naive T$_{regs}$, we isolated CD45RO$^{neg}$ naive T$_{reg}$ cells from tonsils and infected them with HIV in the presence of TLR2 ligand. The frequency of PD-1$^{high}$ cells and AREG expression are much lower in these cultures when compared to non-purified CD4 T cell cultures (compare Supplementary Fig. 13B with Fig. 5A). Also, pre-existing CD45RO$^+$ T$_{reg}$ cells have increased BCL-2 expression compared to CD45RO$^{neg}$ FOXP3$^+$ cells (Supplementary Fig. 13C). Additionally, we also sorted PD-1$^+$T$_{reg}$ and PD-1$^{neg}$ T$_{reg}$ cells from tonsils and examined the expression of secondary markers such as IFN-γ and AREG with and without infection (Supplementary Fig. 14). Consistent with our hypothesis and the results in Fig. 3, purified PD-1$^+$ T$_{reg}$ cells showed higher IFN-γ and AREG expression, compared to PD-1$^{neg}$ T$_{reg}$ cells (Supplementary Fig. 14A, B). They also showed higher expression of Ki-67, HIV-GFP, BCL-2, and FOXP3 than PD-1$^{neg}$ T$_{reg}$ cells (Supplementary Fig. 14C–F). These data support the notion that PD-1$^+$ T$_{regs}$, despite having high infection permissibility, may survive and proliferate better with HIV infection. The intrinsic ability of these cells to survive and proliferate leads to the accumulation of dysfunctional T$_{regs}$. Interestingly, a small proportion of PD-1$^{neg}$ T$_{reg}$ population can also upregulate PD-1 and IFN-γ in the context of HIV infection (but not TLR2 stimulation alone). These cells also showed higher proliferation with TLR2 and IL-1β stimulation in the context of HIV infection but not as much as purified PD-1$^+$ T$_{regs}$ (Supplementary Fig. 14C). Taken together, these data show that, while pre-existing PD-1$^+$FOXP3$^+$ cells might contribute more to the accumulation of dysfunctional T$_{regs}$, naive PD-1$^{neg}$FOXP3$^+$ cells can also be induced to become PD-1$^{high}$ cells expressing high levels of IFN-γ and AREG.

Finally, we determined the ability of IL-1 cytokines and TLR2 ligands to activate AKT kinase downstream in the PI-3K pathway. Because of the ability of IL-1 cytokines to upregulate AREG in tissue T$_{regs}$[56,57,63], we also examined AREG expression in FOXP3$^+$ cells. While HIV was able to increase the accumulation of PD-1$^{hi}$IFN-γ$^+$FOXP3$^+$ cells and moderately induce phosphorylation of AKT and AREG expression, TLR2 ligands, IL-1β, and IL-33 significantly enhanced AKT phosphorylation and AREG expression in FOXP3$^+$ cells (Fig. 5C, D and Supplementary Fig. 15). Based on these observations, we next investigated the function of IL-1β-induced AKT1 signaling pathway in promoting PD-1$^{hi}$IFN-γ$^+$FOXP3$^+$ cells and AREG expression in FOXP3$^+$ cells. Both drugs, the inhibitors of IL-1β signaling (Anakinra) and PI-3K/AKT1 (LY294002), significantly reduced the percentage and absolute cell numbers of HIV-induced PD-1$^{hi}$IFN-γ$^+$FOXP3$^+$ cells in tonsil cultures (Fig. 5E and Supplementary Fig. 16). Also, IL-1β and AKT1 inhibition downmodulated AREG expression in FOXP3$^+$ cells (Fig. 5F), suggesting synergistic roles of HIV, TLR2 ligands, and IL-1β in altering FOXP3$^+$ cells in an AKT1-dependent fashion during HIV infection.

**PD-1 signaling stabilizes the expression of FOXP3 and AREG by downmodulating AEP**. In the above experiments, we observed that IL-1β was able to promote PD-1 expression in FOXP3$^+$ cells in a manner dependent on AKT1 activation (Supplementary Figs. 14 and 16; y-axis). This led us to interrogate whether PD-1 signaling directly regulated HIV-induced PD-1$^{hi}$IFN-γ$^+$FOXP3$^+$ cells. PD-1 has been previously shown to modulate AEP, an endo-lysosomal protease implicated in antigen processing and FOXP3 expression[64,65]. Therefore, we further characterized the PD-1$^{high}$FOXP3$^+$ and PD-

1$^{low}$FOXP3$^+$ cells in HIV-infected CD4$^+$ T cell cultures in the presence of Efavirenz added 28 h after infection. Although PD-1$^{high}$FOXP3$^+$ cells had slightly higher expression of AEP, levels of phosphorylated AEP (pAEP), the active form of AEP enzyme, were precipitously lower than in PD-1$^{low}$FOXP3$^+$ cells (Fig. 6A, first two panels). Also, these PD-1$^{high}$FOXP3$^+$ cells had higher expression (higher mean fluorescent intensity (MFI)) of FOXP3 compared to PD-1$^{low}$FOXP3$^+$ cells (Fig. 6A, third panel). Concurrent with their enhanced survival and proliferation, PD-1$^{high}$FOXP3$^+$ cells had elevated expression of BCL-2 and Ki-67 (Fig. 6B). Engaging PD-1 using PD-1 ligand-Fc (PDL-1-Fc) or inhibiting AEP using an inhibitor increased FOXP3 expression in HIV-infected CD4$^+$ cells (Fig. 6C), suggesting that active PD-1 signaling in the context of IL-1β is involved in the stability of FOXP3 expression during HIV infection. PD-L1-Fc and AEP inhibition heightened the frequency and absolute numbers of PD-1$^{hi}$IFN-γ$^+$FOXP3$^+$ cells (Fig. 6D) showing that the PD-1–AEP axis is critical for the survival and proliferation of PD-1$^{hi}$FOXP3$^+$ cells. Moreover, PD-1 engagement and AEP inhibition promoted the expression of AREG in PD-1$^+$FOXP3$^+$ cells (Fig. 6E). Purified PD-1$^{neg}$ cells that were activated and infected as in Supplementary Fig. 14, lose FOXP3, which further confirms that PD-1 is required for Foxp3 retention (Supplementary Fig. 17). Altogether, these results showed that direct PD-1 signaling enhances FOXP3 and AREG expression by inhibiting AEP in the context of IL-1β expression during HIV infection.

**PD-1$^{high}$FOXP3$^+$IFN-γ$^+$ cells from HIV-infected cultures have little or no suppressive activity**. Next, we explored the function of PD-1$^{hi}$IFN-γ$^+$FOXP3$^+$AREG$^{high}$ cells that were induced during HIV infection in vitro and compared them with purified naive CD4$^+$CD25$^+$CD127$^{low}$FOXP3$^+$ cells activated and infected in a similar manner. To this end, we activated and infected tonsillar CD4$^+$ T cells or purified CD4$^+$CD25$^+$CD127$^{low}$FOXP3$^+$ T$_{regs}$ as before (Fig. 7A, top) and analyzed the proportion of PD-1$^{hi}$IFN-γ$^+$ within the FOXP3$^+$ population. Interestingly, purified T$_{regs}$ harbored significantly lower proportions of PD-1$^{hi}$IFN-γ$^+$ cells (Fig. 7A, bottom, and Fig. 7B), suggesting that PD-1$^{hi}$IFN-γ$^+$FOXP3$^+$AREG$^{high}$ cells are derived preferentially from conventional CD4$^+$ T cells and T$_{regs}$ that upregulate and maintain FOXP3 during activation. Next, we purified the PD-1$^{hi}$CD25$^+$ cells from CD4$^+$ cell cultures, which were HIV infected in the presence of Efavirenz, and examined their suppressive activity. Cells purified from these cultures were ~76–88% FOXP3$^+$ and >50% IFN-γ$^+$ positive (Supplementary Fig. 18). We evaluated their ability to suppress the proliferation of CD4$^+$ T cells by co-culturing them with cell-trace-labeled activated tonsil CD4$^+$CD25$^{neg}$ responder T cells (T$_{resp}$) from the same donor, as shown previously[58]. As controls, we had CD4$^+$CD25$^{neg}$ activated alone and in co-cultures with purified T$_{regs}$ that were activated and infected with HIV in the presence of Efavirenz. As expected, purified T$_{regs}$ reduced the frequency of proliferating T$_{resp}$ cells. However, at all time points after activation, PD-1$^{hi}$CD25$^{hi}$FOXP3$^+$ cells did not affect the proliferation of T$_{resp}$ cells in the co-cultures (Fig. 7C, D). Although it is possible that PD-1$^+$ T$_{regs}$ in tonsillar cultures lost their suppressive capacity due to in vitro culture conditions, these data show that PD-1$^{hi}$CD25$^{hi}$FOXP3$^+$ cells induced during HIV infection were dysfunctional in the context of their direct suppression of CD4$^+$ T cell survival or proliferation. Collectively, these data show that PD-1$^{hi}$IFN-γ$^+$FOXP3$^+$AREG$^{high}$ cells derived from HIV-infected cultures do not suppress CD4$^+$ T cells in vitro.

**The abundance of PD-1$^{hi}$CD25$^{hi}$IFN-γ$^+$AREG$^{hi}$FOXP3$^+$ cells correlates with oral mucosal CD4 hyperactivation in oral mucosa of HIV+ patients**. Although oral mucosa of HIV$^+$ patients had a significantly higher frequency of FOXP3$^+$ cells (Figs. 2D, E and 8A), because of the associated inflammatory

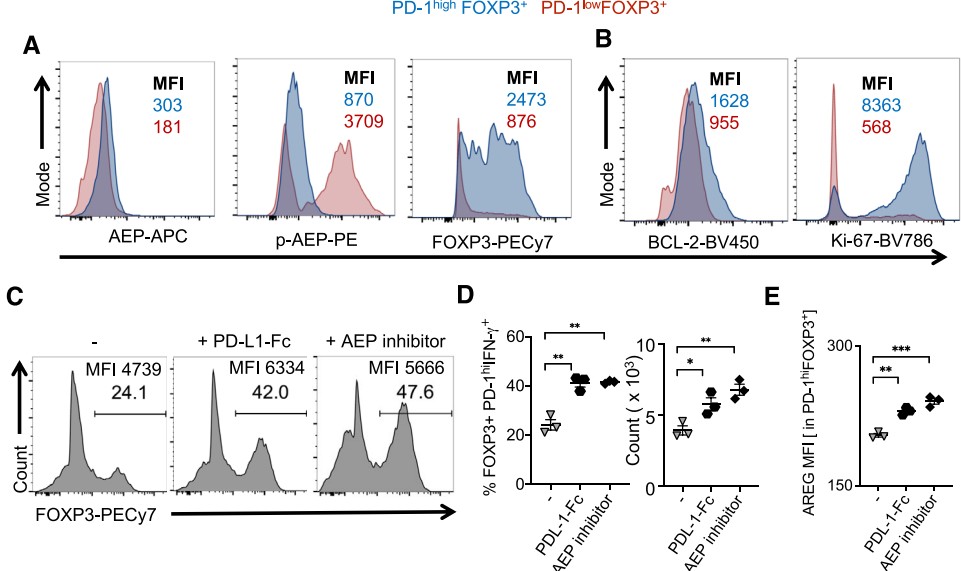

**Fig. 6 PD-1 ligation downmodulates asparaginyl endopeptidase (AEP) and stabilizes the expression of FOXP3 and AREG.** CD4$^+$ T cells were stimulated as in Fig. 5. **A**, **B** AEP, pAEP, FOXP3, BCL-2, and Ki-67 staining in PD-1$^{high}$FOXP3$^+$ (blue) and PD-1$^{low}$FOXP3$^+$ cells 6 days post-infection. Some CD4$^+$ T cells stimulated and infected as above were moved to a plate coated with recombinant human PD-L1/B7-H1 Fc chimera (5 μg/ml) or treated with AEP inhibitor (10 μM) 36 h after infection. Percentage of FOXP3$^+$ cells in CD4$^+$ population and FOXP3 MFI on FOXP3 gated cells (**C**), percentage and absolute cell numbers of PD-1$^{hi}$IFN-γ$^+$ cells in FOXP3$^+$ population (**D**), and AREG expression in FOXP3$^+$ cells (**E**), as determined by flow cytometric analyses. Results represent triplicate experiments with similar results and are presented as mean value +/− SEM (ordinary one-way ANOVA and multiple $t$ tests were conducted; *$P < 0.05$; **<0.005, ***<0.0005).

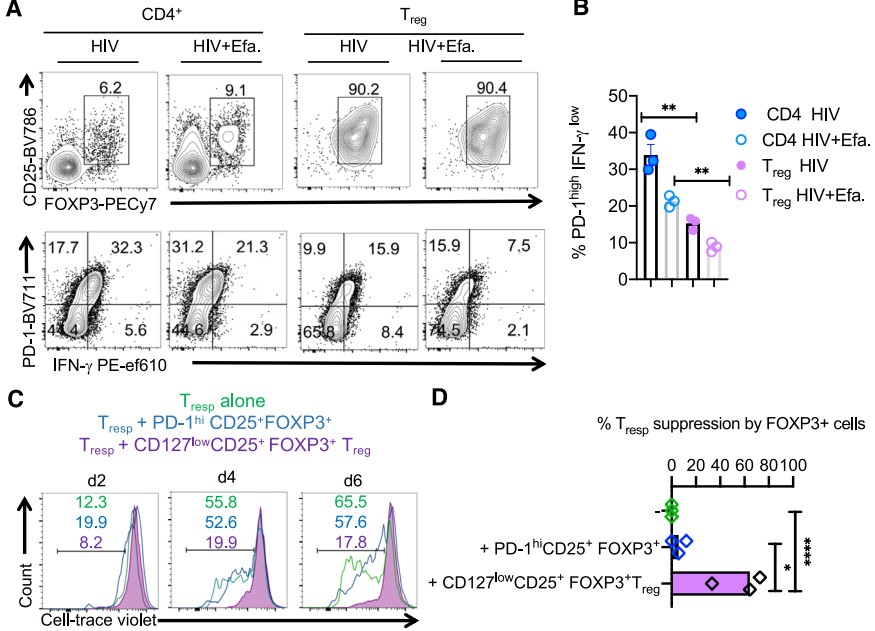

**Fig. 7 PD-1$^+$FOXP3$^+$ cells from HIV-infected cultures have little or no suppressive activity.** Purified CD4$^+$ T cells and T$_{regs}$ were stimulated and infected as in "Methods." **A** CD25 and FOXP3 expression in all cells in the cultures (above) and PD-1 and IFN-γ expression in CD25$^+$FOXP3$^+$ cells (below) at 96 h post infection. **B** Statistical analyses of PD-1$^{hi}$IFN-γ$^+$ cells in FOXP3$^+$ population from these two cultures. **C** PD-1$^{hi}$CD25$^+$ cells were purified from HIV-infected CD4 cultures using sequential sorting of PD-1-PE$^+$ cells and CD25$^{high}$ T$_{reg}$ cells using STEMCELL technology PE isolation and CD25$^+$ T$_{reg}$ isolation kits and were used in co-cultures with cell-trace violet-labeled responder T cells (T$_{resp}$) at ratio 1:1. As controls, T$_{resp}$ cells were cultured alone or co-cultured with purified naive CD127$^{low}$CD25$^+$ T$_{regs}$ that were sequentially sorted to remove CD45RO$^+$CD4$^+$ cells using human CD45RO kit (Miltenyi) and purify CD25$^{high}$ T$_{reg}$ cells using STEMCELL technology CD25$^+$ T$_{reg}$ isolation kits. These control T$_{regs}$ were also previously stimulated and infected the same manner (purple) before co-culture with T$_{resp}$. T$_{resp}$ proliferation was determined by cell-trace dye dilution in PD-1$^{hi}$CD25$^+$ co-culture (blue), control T$_{reg}$ co-cultures (purple), or those cultured alone (green). **D** Statistical analyses of % T$_{resp}$ suppression mean values from three independent experiments showing similar results. **B**, **D** Mean +/− SEM are shown; *$P < 0.05$; **<0.005, ****<0.00005; two-tailed; Mann–Whitney test).

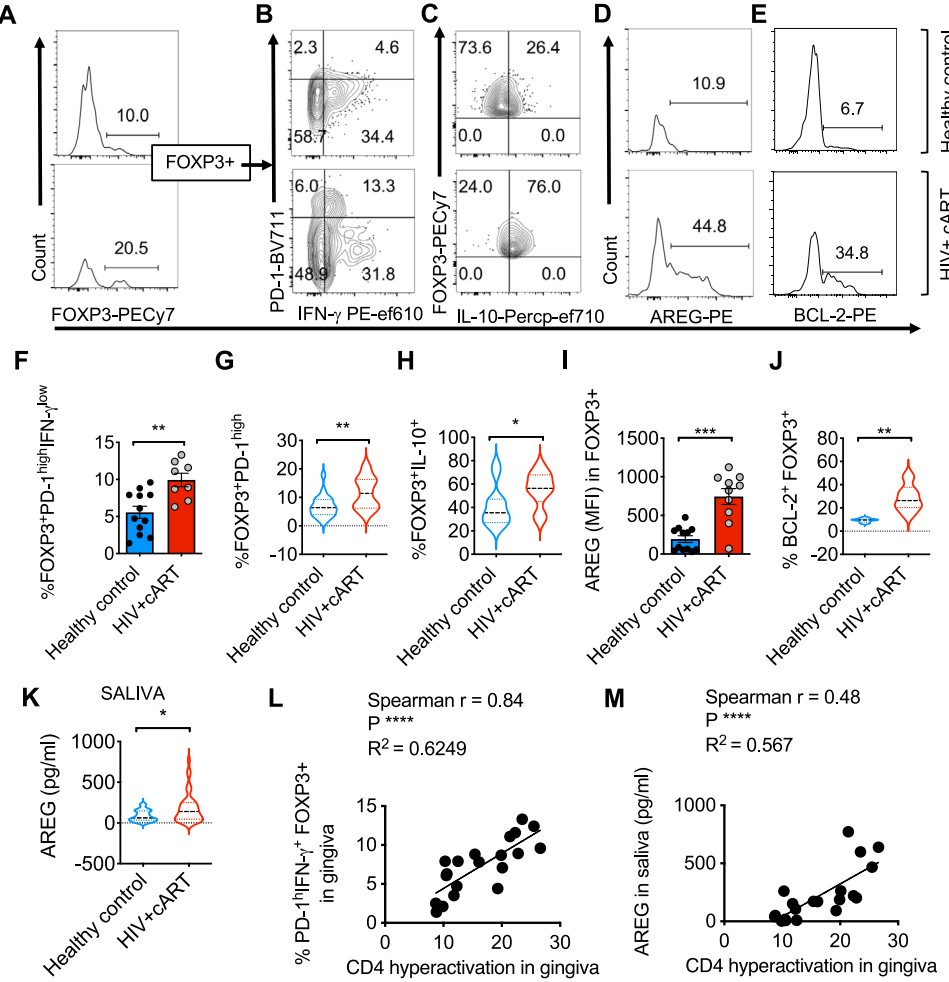

**Fig. 8 HIV$^+$ patients have an increased abundance of PD-1$^{hi}$CD25$^{hi}$IFN-$\gamma^+$AREG$^{hi}$FOXP3$^+$ cells correlating with CD4$^+$ T cell hyperactivation in the oral mucosa.** HOILs from gingival mucosa from healthy controls and HIV$^+$ patients on cART were processed for flow cytometry ex vivo. **A** FOXP3 expression in CD3$^+$CD4$^+$ gated HOIL cells. PD-1 and IFN-$\gamma$ (**B**), IL-10 (**C**), AREG (**D**), and BCL-2 (**E**) expression in FOXP3$^+$ population. Statistical analyses and comparison between the groups for % PD-1$^{hi}$IFN-$\gamma^+$ cells ($n = 20$; **$P = 0.0025$) (**F**), % PD-1$^{hi}$ cells ($n = 35$; **$P = 0.005$) (**G**), % IL-10$^+$ cells ($n = 22$; *$P = 0.024$) (**H**), AREG expression ($n = 22$; ***$P = 0.0002$) (**I**), and BCL-2 expression ($n = 12$; **$P = 0.0022$) (**J**) in FOXP3$^+$ population. **K** ELISA quantification of AREG levels in saliva ($n = 78$; *$P < 0.03$). **L, M** Correlation of % PD-1$^{hi}$CD25$^+$ cells in FOXP3$^+$ population (**L**) and salivary AREG (**M**), with effector CD4 hyperactivation (% CD38$^+$HLADR$^+$ in FOXP3$^{neg}$CD4$^+$ T cells in gingival mucosa; Fig. 1E; $n = 20$). **F–K** $P$ values two-tailed; Mann–Whitney test; each data point represents a study participant, and the data are presented as mean value +/− SEM. Source data are provided as a Source data file.

signature and elevated IL-1β signaling (Figs. 1D–F and 2A), we hypothesized that FOXP3$^+$ cells accumulating in oral mucosa of HIV$^+$ patients may also be dysfunctional. To test this notion, we evaluated the expression of dysfunctional markers, PD-1 and IFN-γ. Although only a small proportion (~6–9%) of CD4$^+$CD25$^+$FOXP3$^+$ cells from healthy controls expressed PD-1, about 14–19% of them expressed PD-1 in HIV$^+$ patients (Fig. 8B, $y$-axis, and Fig. 8E, F). Concurring with the results from oral MALT HIV infection in vitro (Fig. 3), HIV$^+$ patients on cART also had a significantly higher percentage of PD-1$^{high}$-FOXP3$^+$ cells, co-expressing IFN-γ, IL-10, and elevated expression of AREG and BCL-2 in the oral mucosa (Fig. 8B, C, upper right quadrants, Fig. 8D–J, and Supplementary Fig. 19; FMO controls). AREG levels in saliva were also found to be elevated in HIV$^+$ patients on cART when compared to healthy control individuals (Fig. 8K). These FOXP3$^+$ cells fit the profile of dysfunctional FOXP3$^+$ cells incapable of CD4$^+$ T cell suppression in vitro (Fig. 7). While we could not directly evaluate the suppressive function of patient oral mucosal FOXP3$^+$ cells because of the inability to obtain enough purified T$_{regs}$ required for the assay, the frequencies of PD-1$^+$IFN-γ$^+$FOXP3$^+$ cells and salivary

AREG levels showed a significant positive correlation with CD4$^+$ hyperactivation (CD38 and HLA-DR co-expression) in the oral mucosa (Fig. 8L, M). These data suggest that FOXP3$^+$ cells in HIV$^+$ patients might indeed be impaired in their ability to suppress CD4$^+$ T cells in the oral mucosa. Taken together, these data from oral gingival mucosal cells of PLWH substantiate the results from in vitro tonsil experiments and demonstrate that dysfunctional PD-1$^+$AREG$^+$FOXP3$^+$ cells strongly correlate with CD4 hyperactivation and contribute to the dysregulated immune landscape in treated HIV$^+$ patients.

## Discussion

**A unique population of tissue T$_{reg}$-like FOXP3$^+$ cells accumulates in oral MALT and mucosal tissue during HIV infection.** Immunological complications in HIV$^+$ individuals on treatment appear to result from a self-perpetuating cycle of events involving microbial translocation, excessive release of pro-inflammatory cytokines, and CD4 T cell activation, which, in excess, increases the cellular targets for HIV infection and subsequent immune exhaustion[12]. Here we show that HIV and TLR2

ligands can lead to the accumulation of non-suppressive PD-1$^{high}$IFN-γ$^+$AREG$^+$FOXP3$^+$ cells in an IL-1β-dependent manner. These data support the notion that PD-1$^+$ T$_{regs}$ may intrinsically survive and proliferate better with HIV infection leading to the accumulation of dysfunctional T$_{regs}$. Interestingly a small proportion of PD-1$^{neg}$ T$_{reg}$ or CD45RO$^{neg}$ naive T$_{reg}$ cells can also upregulate PD-1 and IFN-γ in the context of HIV infection (and not TLR2 stimulation alone; Fig. 5A, see Uninfected), which also show higher proliferation with TLR2 and IL-1β stimulation in the context of HIV infection (Supplementary Figs. 13 and 14). It is conceivable that, in mucosal tissues and tonsils that are known to be enriched with pre-existing PD-1$^+$FOXP3$^+$ and memory FOXP3$^+$ cells, these cells may contribute better to the accrual of dysfunctional T$_{reg}$ cells than naive T cells. However, the local induction from PD-1$^{neg}$FOXP3$^+$ cells and recently activated FOXP3$^+$ cells cannot be ruled out in the process. Therefore, we speculate that, in HIV$^+$ individuals, the effects of HIV, TLR2 activation, and IL-1β on FOXP3$^+$ cell accumulation are driven by complementary and synergistic processes of induction, survival, and proliferation of FOXP3$^+$ cells. By demonstrating the mechanistic details by which AKT1 and PD-1 enhance FOXP3 stability and expansion of dysfunctional FOXP3$^+$ cells, our study unveils a critical process that may contribute to HIV-mediated CD4$^+$ T cell activation that persists in oral mucosa during therapy.

Our study is consistent with previous data showing HIV-1-driven T$_{reg}$ accumulation in lymphoid tissues and the association of TLR2 ligands and the NLRP3 inflammasome in immune activation and disease progression in HIV/acquired immunodeficiency syndrome (AIDS)[31,66,67]. Elevated levels of soluble salivary TLR2 ligands and CD14 levels in conjunction with higher TLR and inflammasome activation (Fig. 2A, B) suggest that these features might be strongly linked to the previously established dysbiosis of the oral microbiome in HIV$^+$ patients[39]. While the additional role of endocytosed HIV and the resultant TLR7/8 activation cannot be ruled out[68], our data show mechanistic details by which TLR2 ligands and IL-1β/inflammasome might contribute to proliferation and dysfunction of FOXP3$^+$ cells and excessive immune activation in the oral mucosa. The question that remains to be addressed is whether the dysfunction of non-regulatory T cells, and perhaps other cell types, is due to HIV, altered T$_{reg}$ function, or both.

The human tonsil infection model that we employed here has been previously shown to support productive HIV infection, cell death, and release of cytokines such as IL-1β and IL-10, which is similar to an HIV-induced inflammation in humans[34,50,69,70]. Also, being an oral MALT system, it provides a preview into immune cell alterations in oral mucosal tissues. Because the rationale of the study was to determine the underlying oral mucosal dysregulation in HIV+ cART patients, we wanted to mimic the effect of HIV in the presence and absence of HIV inhibitor. Considering the different antiviral regimens that patients take, our in vitro experiments may not be exactly physiologically relevant but may show similarity to HIV+ cART patient samples. With this system, we show that FOXP3$^+$ cells are highly permissible to HIV infection and undergo cell death during HIV infection (Figs. 3C and 4B). However, a unique population of PD-1$^{high}$IFN-γ$^+$AREG$^{high}$FOXP3$^+$ cells that expressed anti-apoptotic BCL-2 and are rescued from cell death appeared in cultures, thus unveiling new features of FOXP3$^+$ cell dysregulation during HIV infection. These cells expressing CD45RO, IL-1R, and ST-2 have an activated/memory T$_{reg}$-like phenotype[63,71] and proliferated with TLR2 agonists and IL-1β even in the presence of HIV reverse transcriptase inhibition (Fig. 5). These PD-1$^{hi}$ T$_{regs}$ responding to IL-1β and expressing AREG resemble the tissue T$_{regs}$ induced by IL-33, another IL-1

superfamily cytokine. Similar to tissue T$_{regs}$, they may have non-suppressive roles and may function toward mucosal tissue repair during inflammation. They might also differentially govern non-immunological processes in oral mucosa of HIV$^+$ patients, compatible with previously described functions of tissue T$_{regs}$[57]. PD-1$^{high}$FOXP3$^+$ cells appeared to have low MFI of IFN-γ expression (Figs. 3D and 4C and Supplementary Fig. 16), consistent with FOXP3 and PD-1-mediated inhibition of IFN-γ[59]. However, these cells from few other experiments showed higher MFI of IFN-γ expression (Supplementary Fig. 15). The reason behind these discordant results is unclear but is likely linked to the differences in donors and their underlying tonsillitis. IL-1β enhanced PD-1 upregulation as well as the proliferation of PD-1$^{high}$IFN-γ$^+$AREG$^{high}$FOXP3$^+$ cells in a PI-3K/AKT-dependent manner (Fig. 5E and Supplementary Fig. 16). Despite the proliferation and enhanced FOXP3 protein stability conferred by PD-1, these PD-1$^{high}$IFN-γ$^+$AREG$^{high}$FOXP3$^+$ cells that are generated during HIV infection lack suppressive ability in blocking CD4 T cell proliferation. This finding concurs with previous data showing cytokines that can activate PI-3K/AKT function or maintain T cell responsiveness to IL-2 in CD4$^+$ cells can also abrogate T$_{reg}$ suppression[58–60,72]. These data are also in accordance with the previous studies implicating heightened PD-1 and IFN-γ expression in T$_{reg}$ dysfunction[73–75].

BLIMP1 is a transcriptional repressor that is critical for IL-10 expression in TFR cells. TFR cells regulate B cells and TFH cells, thereby controlling the germinal center response, autoantibody production, and autoimmune destruction[53]. Whereas BLIMP1 is expressed in a proportion of lymphoid FOXP3$^+$ cells, it is expressed in a majority of FOXP3$^+$ cells found in gut mucosa and tissues and is likely crucial for their IL-10 expression in environmental interfaces[76]. Whether the PD-1$^{high}$IFN-γ$^+$AREG$^{high}$FOXP3$^+$ cells in HIV$^+$ oral mucosa that express IL-10 also co-express BLIMP-1 is an important question that remains to be investigated in the future. Although AREG was originally described as an epithelial cell-derived factor and is a member of the epidermal growth factor receptor family, it is now clear that this protein can be expressed by activated immune cells during inflammatory conditions[56]. AREG produced by T cells is critical for type 2 adaptive immune responses and gut epithelial cell proliferation that facilitates helminth parasite clearance. Tissue T$_{regs}$ are shown to express this cytokine and are critical for non-immunological tissue repair functions[57]. Here we show that AREG expression is high in PD-1$^{high}$FOXP3$^+$ cells and is further upregulated by IL-1β in PI-3K/AKT-dependent manner. Alarmins such as IL-18 and IL-33 have been previously shown to upregulate this cytokine in FOXP3$^+$ cells[57]. In a tonsillar CD4$^+$ T cell environment, although HIV did not induce IL-33 expression (Supplementary Fig. 11), the exogenous addition of both IL-1β and IL-33 upregulated AREG expression in FOXP3$^+$ cells (Fig. 5D). However, in the context of HIV infection, only the endogenous IL-1β released due to caspase-1 activity upregulated AREG in vitro (Fig. 5B). PD-1 enhancement and AEP inhibition due to IL-1β upregulates AREG expression in PD-1$^{high}$FOXP3$^+$ cells. The mechanism underlying the inverse relationship between AEP activity and AREG expression remains to be explored in the future.

**In vivo evidence for the enrichment of PD-1$^{high}$IFN-γ$^+$AREG$^{high}$FOXP3$^+$ cells in oral mucosa of HIV+ individuals on therapy.** Our results from in vitro HIV infection experiments support transcriptomic and flow cytometric profiling results from HIV$^+$ patients undergoing suppressive antiviral therapy, whose oral mucosa also revealed TLR signaling pathway upregulation and an inflammasome gene signature that paralleled excessive CD4$^+$ T cell

activation and enrichment of PD-1$^{high}$IFN-γ$^+$AREG$^{high}$FOXP3$^+$ cells (Figs. 1 and 8). We found distinct populations of CD38$^+$HLA-DR$^+$ in CD4$^+$ T cells in oral mucosa (Fig. 2D). It may be related to the downregulation of HLA-DR (major histocompatibility complex) in CD4$^+$ T cells. While HLA-DR downregulation in monocytes is associated with immune suppression, it remains to be seen if this is the case for CD4$^+$ T cells. PD-1$^{high}$IFN-γ$^+$AREG$^{high}$FOXP3$^+$ cells we describe here resemble the dysfunctional Th1 T$_{regs}$, which also display constitutive activation of PI3K/AKT/Foxo1/3 signaling cascade in multiple sclerosis patients[77]. However, we also show that CD4$^+$ T cell hyperactivation and enrichment of PD-1$^{high}$IFN-γ$^+$AREG$^{high}$FOXP3$^+$ cells in the oral mucosa of HIV$^+$ patients coincide with increased TLR2 signaling and salivary s-TLR2 ligands (Figs. 1B, D and 2B). We speculate that increased s-TLR2 ligands we observed in HIV$^+$ patients (Fig. 2B) maybe due to TLR2 shedding as a consequence of increased pro-inflammatory signaling downstream to the TLR2 signaling[78]. PD-1 expression on CD4$^+$ T cells and T$_{regs}$ is known to be associated with immune activation as well as HIV$^+$ reservoirs, and thus this molecule is targeted for therapy in HIV$^+$ patients[74,79,80]. Gut mucosa, a tissue enriched with lymphoid structures and bombarded by microbial products as a result of microbial translocation, serves as the largest reservoir[81]. Survival advantage and proliferation of CD4$^+$ T cells by homeostatic cytokines and chronic exposure to antigens or other stimulants contribute to the expansion of latently infected cells and consequent establishment of reservoirs[81]. Therefore, considering that HIV$^+$ patients on therapy show oral microbiome dysbiosis[38,39], the tissue T$_{reg}$-like PD-1$^{high}$FOXP3$^+$ cells appear to fit these criteria and might provide a supportive environment for the maintenance of HIV reservoirs in the oral mucosa. This tenet is consistent with previous reports showing that FOXP3$^+$ cells are highly permissible and contribute to latent reservoir compartments[82,83]. Future studies are required to conclusively verify this possibility in the oral mucosal environment. While the in vivo relevance of impaired suppressive capacity of PD-1$^{high}$IFN-γ$^+$AREG$^{high}$FOXP3$^+$ cells in HIV-infected cultures is unclear (Fig. 7), these T$_{reg}$ cells with high expression of BCL-2 and KI-67 and resistance to apoptosis (Figs. 3E and 4D) suggest that they may be long lived and may undergo continuous cycling. This is consistent with previous reports showing T$_{regs}$' resistance to apoptosis[84,85]. The phenotypic resemblance of these non-suppressive FOXP3$^+$ cells with PD-1$^{high}$IFN-γ$^+$AREG$^{high}$FOXP3$^+$ cells in the oral mucosa of HIV+ cART patients, correlating with CD4 hyperactivation (Fig. 8), would imply that these might be long lived and dysfunctional in HIV+ patients. However, we cannot rule out that these cells co-expressing IL-10 may still inhibit CD8$^+$ T cells, myeloid populations, neutrophils, and resident macrophages, providing an immune-suppressive environment. Taken together, these results show that persistent microbial stimulants and excessive IL-1β signaling perturb FOXP3$^+$ T cell homeostasis and function and underlie the processes of residual oral mucosal immune dysfunction in HIV$^+$ patients on therapy.

## Methods

**Human PBMCs, gingival biopsies, and tonsils.** Human blood, gingival biopsies, and saliva were obtained with informed consents from healthy individuals and Cleveland HIV$^+$ cohort under a protocol approved by the University Hospitals Cleveland Medical Center Institutional Review Board. Healthy control subjects are at least 18 years of age and in good general health (Supplementary Table 1). Exclusion criteria were oral inflammatory lesions (including gingivitis and periodontitis), oral cancer diagnosis, soft tissue lesions, and the use of tobacco in the past month. HIV$^+$ participants were ≥18 years and were HIV positive with cART treatment for at least 1 year. More than 75% of HIV$^+$ patients reported prior and current soft tissue lesions, gingivitis, and periodontitis. Exclusion criteria were oral cancer diagnosis and the use of tobacco in the past month. The absence of tobacco use was confirmed by Cotinine ELISA in saliva. CD4$^+$ counts were at least 350–700/μl for the control and HIV$^+$ patients. Palatine tonsils were obtained as discards from tonsillectomy surgeries performed at University Hospitals Cleveland Medical Center through the Histology Tissue Procurement Facility following an Institutional Review Board-approved protocol. PBMCs were collected from blood using Ficoll-Paque PLUS centrifugation and subsequent washing with phosphate-buffered saline (PBS). A single-cell suspension of gingival tissues and tonsils were prepared by Collagenase 1A digestion and processed for flow cytometry or cell culture.

**HIV infection in vitro.** HIV infections in tonsil cultures were performed using X4-tropic NL43-GFP-IRES-Nef or HIV-NLGNef, a recombinant virus with NL4-3 backbone expressing GFP and Nef on a bicistronic transcript[47,86]. Viral constructs were obtained through NIH AIDS Reagent Program and the viruses were generated by transfecting 293T cells with proviral DNA. The R5-tropic virus was created replacing the Env in NL43-GFP-IRES-Nef with the EcoR1-Bam fragment from NLAD8, an NL43 construct containing CCR5-tropic HIV-1 ADA envelope[47,48]. Concentrated virus stock titers were determined by p24 ELISA. For infections, tonsil tissues were digested using collagenase, and a single-cell suspension of the human tonsillar culture (HTC) (1 million cells/well) was plated with α-CD3 (1 μg/ml) and α-CD28 (1 μg/ml) TCR-activating antibodies in U-bottom 96-well plates at least in triplicate wells. After 48 h, the bulk HTC were spinoculated with replication-competent HIV-1 NLAD8-GFP virus stock (70 ng of p24/10$^6$ cells). Cells were rested for 48 h in medium without TCR activation in select experiments. As indicated in some experiments, purified CD4$^+$ T cells and T$_{regs}$ were used instead of whole HTC. Twenty-four-to-36 h post-infection, 50% of the cells and media were removed and replaced with media containing fresh media, Efavirenz, and the indicated cytokines or reagents. This allowed an initial round of infection and cell death to occur before the addition of the indicated reagents. When indicated, transforming growth factor (TGF)-β1 (10 ng/ml) and IL-2 (100 U/ml) were also added during this time to induce and maintain FOXP3$^+$ cells. Confirmatory experiments were performed using both X4- and R5-tropic viruses[86]. Cells were cultured in complete RPMI-1640 (Hyclone) supplemented with 10% human serum, 100 U/ml penicillin, 100 μg/ml streptomycin, 2 mM glutamine, 10 mM HEPES, and 1 mM sodium pyruvate. To determine productive HIV infection (GFP) and regulation of protein expression, cells were analyzed by flow cytometry on day 2–8 post-infection. Flow cytometric analyses and ELISAs were performed in triplicates using tonsils from at least three independent donors.

**Antibodies and reagents.** Unconjugated or fluorochrome-conjugated antibodies for human CD28 (CD28.2), CD25 (M-A251), CD4 (OKT4), CD45 (HI30), CD8 (RPA-T8), HLA-DR (LN3), IFN-γ (4S.B3), IL-17A (eBio64DEC17), FOXP3 (236A/E7), Phospho-AKT 1 (Ser473) (SDRNR), BCL-6 (BCL-UP), CXCR5 (MU5UBEE), Ki-67 (SolA15), IL-10 (JES3-9D7), AREG (AREG559), IL-6 (MQ2-13A5), ST2 (goat polyclonal), phospho-caspase 1 (polyclonal), LY294002, and cell-trace violet were all purchased from Thermofisher Scientific. CD279 (PD-1) (EH12.1), CXCR4 (12g5), CCR5 (2D7/CCR5), BCL-2 (Bcl-2/100), CD19 (SJ25C1), CD38 (HIT2), CD3 (HIT3a), and IL-1R1 (hIL1R-M1) were from BD Biosciences. Phospho-AEP (SER 226) antibody, Efavirenz (SML1284-1ML), and AEP inhibitor were from Millipore Sigma. Biotinylated antibody for AEP, BLIMP1 antibody (646702), Human TGF-β1, and the chimeric PDL-1-Fc were purchased from R&D Systems. The primary antibodies used for surface and intracellular staining were used at 1:200 and 1:50 dilutions, respectively, or according to the manufacturer's recommended dilution. IFN-γ, IL-1R1, and biotinylated antibody for AEP antibodies were used at 1:100 dilution. Appropriate secondary antibodies such as secondary donkey anti-mouse IgG-BV421 (for IL1-RI staining), anti-goat IgG (H + L) superclonal™-Alexa Fluor 647 (for ST2 staining) streptavidin-APC, and anti-rabbit PE or APC antibodies were purchased from Jackson Immunoresearch or Invitrogen/Thermofisher. The secondary antibodies for flow cytometry were used at a dilution of 1:500. Anti-Biotin multi-sort and human CD4$^+$CD45RO$^+$ isolation kits were purchased from Miltenyi Biotec (Auburn, CA). PE$^+$ cell, CD4$^+$ T cell, and T$_{reg}$ isolation kits were purchased from Stem Cell Technologies (Vancouver, Canada). Recombinant IL-2, IL-1β, and IL-33 cytokines were purchased from BioBasic Inc. (Amherst, NY). s-CD14, s-TLR2, Cotinine, AREG, IL-1β, and IL-6 ELISA kits were from Boster Bio (Pleasanton, CA). IL-1 receptor antagonist Anakinra was a kind gift from Dr. Su at NIAID, NIH. Nigericin and PG-LPS were purchased from Invivogen. Heat-killed *Candida albicans* germ tubes (HKGT) were prepared in the laboratory by growing the blastospores (10$^9$/ml) into germ tubes in complete RPMI at 37 °C for 4–6 h and heat killing the germ tubes at 75 °C for 60 min.

**Fluorochrome antibody staining and flow cytometry.** For single-cell flow cytometric analyses, surface receptors were first stained using the antibodies in PBS/bovine serum albumin. Live-Dead viability staining was performed to detect and remove dead cells in the analyses. For FOXP3 and other intracellular protein stainings, the cells were fixed with a FOXP3 fixation–permeabilization set (Thermofisher Scientific) after the surface staining. Unstimulated, un-stain, isotype, secondary antibody alone, single stain, and FMO controls were included in all the preliminary and confirmatory experiments, and appropriate controls were chosen. Before intracellular cytokine staining, cultures were re-stimulated with phorbol 12-myristate 13-acetate (50 ng/ml) and Ionomycin (500 ng/ml) for 4 h, with brefeldin-A (10 μg/ml) added in the last 2 h. For p-AKT1 staining, the cells were washed, fixed, and stained with a Phosflow Staining Kit (BD Biosciences) using the manufacturer's protocol. Data were acquired using BD Fortessa cytometers (BD FACSDiva software ver.7) and analyzed using the FlowJo 9.8–10.7.1 software versions. Populations were pre-gated for lymphocyte, singlet, viable, CD3$^+$, and CD8$^-$ or CD4$^+$ cells during flow cytometric analyses, unless otherwise specified.

**PD-1 engagement and Treg suppression assay in vitro**. Tonsil cells were stimulated and infected in U-bottom 96-well plates as above. Thirty-six hours after infection, the cells were moved to the plate coated with PDL-1-Fc for PD-1 engagement. The plates were previously coated with PDL-1-Fc for 12–16 h. Appropriate isotype control (IgG2a) was used in control wells. Flow cytometry was performed on day 4 or 5 after PD-1 engagement. For the suppression assay, three groups of magnetic sorted cells purified ex vivo from tonsils were activated with CD3/CD28 antibodies with added TGF-β1 and IL-2 for 96 h: (I) CD45RO$^{neg}$ naive CD4$^+$CD25$^+$CD127$^{low}$ T$_{regs}$ (>90% FOXP3$^+$), (II) purified CD4$^+$CD25$^-$ T cells that were subsequently used as responder T cells (T$_{resp}$) in the co-culture assay, and (III) purified CD4$^+$ T cells infected with HIV. PD-1$^{high}$CD25$^+$ cells purified from these cultures were 80–88% FOXP3$^+$, IFN-γ$^+$ (52%) (Supplementary Fig. 18), AREG$^{high}$ and were used as PD-1$^{high}$CD25$^+$FOXP3$^+$ cells. For co-culture T$_{reg}$ suppression assay, $3 \times 10^4$ T$_{resp}$ cells were labeled with cell-trace violet and co-cultured with $3 \times 10^4$ CD4$^+$CD25$^+$CD127$^{low}$ T$_{regs}$ or $3 \times 10^4$ PD-1$^{high}$CD25$^+$ cells in triplicate wells of U-bottomed 96-well plates in the presence of soluble 1 µg/ml α-CD3 and 1 µg/ml α-CD28 antibodies for the indicated duration[58].

**RNA sequencing**. Sample preparation, sequencing, alignment, and data analyses were performed by Novogene genomic services. Strand-specific whole-transcriptome sequencing libraries were prepared using the NEB Next® Ultra™ RNA Library Prep Kit. The sequencing used a paired-end protocol (PE150). Indexed RNA-seq libraries were sequenced on a HiSeq2500 with Illumina TruSeq V4 chemistry (Illumina, San Diego, CA, USA). The FASTQ files with 125 bp paired-end reads were processed using Trimmomatic (version 0.30) to remove adapter sequences. The trimmed FASTQ data were aligned to the human genome with STAR (version 2.5), which used GENCODE gtf file version 4 (Ensembl 78).

*Differential expression analysis.* The gene reads count data from HOIL and PBMC samples, each derived from three independent human individuals were normalized with R Package limma (version 3.26.8) and analyzed with an unpaired $t$ test. HOIL samples from three control individuals were pooled and compared with three independent HIV+ individuals. The normalized reads count data were used to generate RPKM values for the heatmap display.

*Pathway analysis and heatmaps.* The differentially expressed gene list was generated using unbiased molecular and cellular functional analyses. Heatmaps for different cytokine signatures were created in R using the heatmap.2 function in g plots (version 2.17.0). GSEA was performed using the GSEA software obtained from the Broad Institute (http://www.broad.mit.edu/GSEA). REACTOME, GO, and MSigDB gene sets and reference pathways were employed when relevant. The whole gene list was ranked before uploading to the GSEA software for pathway analysis. Normalized maximum deviation from zero was recorded as the enrichment score and normalized for obtaining normalized enrichment score.

**Statistical analyses**. *P* values were calculated by Mann–Whitney test in Prism 8 (GraphPad Software, Inc.) assuming random distribution unless otherwise noted. For some multiple comparisons within in vitro culture groups, one-way analysis of variance (ANOVA) was used. Unpaired $t$ test and two-way ANOVA were used for multiple comparisons between two or more groups. Bonferroni $t$ test was the post hoc test used for multiple comparisons. *$P$ < 0.05 were considered significant. To measure the strength of the association, correlation plots, Spearman ($r$), and simple linear regression analyses ($R^2$) were used, and an alpha value of *<0.05 was considered significant.

**Reporting summary**. Further information on research design is available in the Nature Research Reporting Summary linked to this article.

## Data availability
RNA sequencing data from healthy human participants that support the findings of this study have been deposited in GEO, NCBI with the GSE167211 accession code. Transcriptome data from HIV+ patients are deposited at the NCBI Genotypes and Phenotypes (dbGaP) data repository. These data are open to general research use (dbGaP Study Accession: phs002364.v1.p1). Other data that support the findings of this study are also available from the corresponding author upon reasonable request. Source data are provided with this paper.

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

## Acknowledgements

P.P. was supported by CWRU Center for AIDS Research (CFAR) Catalytic award and RO1DE026923 NIH/NIDCR funding. We thank Jennifer Bongorno-Hurt for preparing HIV viral stocks and Sangeetha Jayaraman for technical assistance with patient recruitment paperwork and assessing data in a masked fashion. We acknowledge Rafick P Sekaly for critically reading the manuscript and valuable suggestions. We thank Ms. Patricia Mehosky for proof-reading the manuscript.

## Author contributions

P.P. designed the study, performed experiments, analyzed data, supervised the project, and wrote the manuscript. F.F. and A.P.d.S. provided gingival biopsies from human participants, and R.A. referred the patients to the study. N.B. and E.S. performed the experiments, analyzed ELISA data, and contributed to discussions. E.S. obtained consents from patients, collected the saliva and blood, and performed ELISA. A.T. provided statistical analysis consultation for bioinformatics data. D.M., A.D.L., N.G., J.K., and M.M.L. read the manuscript and contributed to discussions.

## Competing interests

The authors declare no competing interests.
