## [Peer Review File · Nature Communications]

REVIEWER COMMENTS

Reviewer #2 (Remarks to the Author):

In the manuscript, "PD-1 dependent expansion of Amphiregulin+FoxP3+ cells is associated with oral immune dysfunction in HIV patients on therapy" by Bhaskaran et al., the authors characterized a novel CD4 T-cell population they exclusively find the tonsils of HIV patients under anti-viral therapy. This T-cell population shows all the hallmarks of regulatory T-cells (Tregs) as these cells express high levels of CD25 and FoxP3 combined with a low expression of CD127. In addition, this T-cell population expresses the activation marker PD-1 and the cytokines IFN γ , IL-10 and Amphiregulin. Strikingly, the frequency of this T-cell population directly correlates with a general hyperactivation state (based on CD38 and MHC-II expression) of effector CD4 T-cells. The experiments are all performed in a technical sound way and the presented data appear quite intriguing.

Unfortunately, all presented data remain purely descriptive in kind. Thus, it remains doubtful whether the conclusions of the manuscript are justified.

For instance, it is not clear whether an inflammatory environment causes the phenotype of these cells or whether the presence of these cells causes an inflammatory environment. That inflammatory cytokines lead to a positive feedback loop via the suppression of Treg function is well established and not specific for HIV. For instance, TLR-2 mediated Treg activation induces Treg proliferation and diminishes their suppressive capacity (Sutmoller et al. 2006), as the authors demonstrate in their manuscript.

Thus, in order to verify their conclusions (Line 445 – 446), it might be important that the authors isolate "naïve"/CD62L expressing Tregs and determine whether indeed TLR-2 ligand mediated activation of Tregs combined with IL-1 induces their differentiation into Amphiregulin expressing Tregs. If so, then this could be reapplied to the situation in HIV patients.

Minor comments:

- To measure the suppressive capacity of Tregs after an extended in vitro culture (four days) and HIV infection appears disingenuous as such culture conditions in no way reflect the phenotype of these cells directly ex vivo.
- Amphiregulin and T2 (the IL-33R) expression is typically associated with tissue resident Tregs. This aspect should be discussed in the manuscript.
- PMA / Ionomycin is not a physiological stimulus for Tregs. Are these Tregs also producing pro-inflammatory cytokines, such as IL-1 and IFN γ , following TCR mediated activation, as the authors performed in Figure 3.
- Figure 3a – how do you get FoxP3 negative Tregs..?

Reviewer #3 (Remarks to the Author):

This manuscript by Bhaskaran et al identified the presence of a population of Tregs that PD-1, IFN γ , Amphiregulin and IL-10. These cells were notable in that they were incapable of suppressing CD4+ T cells in vitro. A major concern is the conclusion regarding induction of cells. This is not formally shown as populations are not sorted to definitively show the differentiation of one type of cell into another. Additionally, in several places there is a comparison with PBMC although the rationale for this and the significance is not clear beyond Fig. 1 and is a distraction from the main point of the study. Finally, in the absence of the ability to demonstrate reduced suppressive activity in the cells from patients on cART, whether the in vitro results, while intriguing, are the case in HIV+ individuals is speculative. Based on the in vitro studies, it would seem that the model is that infection early on results in the presence of ineffective Tregs that is maintained for long periods of time. Is there an indication that these cells could be long lived in a way that would fit

this model? While challenging to prove, the authors should discuss this or clearly define the model if this is not the case. Also it is not clear if the act of infection or some other effect of the infection culture is responsible for the loss of suppressive activity (Fig. 7) as infected Tregs vs non-infected Tregs are not assessed. Finally, a number of the data shown appear to confirm those previously published and as presented the studies are at times difficult to follow. These issues significantly weaken the impact of the study. In addition the following are concerns are noted.

1. From the methods it appears that >75% of the HIV+ individuals in Fig.1 had/have infection/inflammatory conditions independent of HIV. Wouldn't this contribute to the increase inflammation? Wouldn't a similar set of non-HIV+ individuals be the appropriate comparator?
2. There is some question as to the gating for positivity. At time it looks as if the gate runs through the middle of a population (CD38 in 1E) or includes negative cells (FoxP3 in Fig 4A). In Fig S8 why is the cut-off for IFN γ so high if it based on the FMO control? Perhaps a better approach would be the non-restimulated cells which take into account the non-specific staining of the antibody. Also the authors should use caution when it appears the overall background of a sample is higher than another as is the case for 2D (apparent MFI of the double negative population tope panel) as it may skew the results.
3. The two very distinct populations present in the HIV+cART CD38+DR+ gate merits comment.
4. It is not clear why the authors state IL-6 and IL-1 are too low to detect when they show data for these measurements in S3.
5. The Suppl. Figures would benefit from expanded legends that include the experimental detail as opposed to referring back to other figures, including the number tested.
6. The CD14 data in 2B are not described in the results section.
7. The authors conclude there are alterations in TLR-2 signaling (Line 149), although this is not directly shown.
8. Statistical analysis using multiple samples would be required to make statements of significance in lines 192-95.
9. The conclusion that memory Tregs are the most permissible would seem to require sorting and comparison of this population and CD45RO negative cells.
10. It is not clear what led to the statement "They also suggest that HIV infection may alter the functionality of FOXP3+ cells in oral mucosal lymphoid compartments." as no functional data have been shown. line 215
11. The very low level of IFN γ in the FoxP3+ cells merits noting in the results.
12. It was surprising that the authors did not sort PD-1 high versus low TFR and infect the cultures to address the interrelationship between these two populations with regard to infection and the bcl-2 expression and susceptibility to death prior to and after infection. This would provide definitive data regarding the PD-1high population.
13. Once the PD-1hi,CD25, CXCR5,AMREG, CD127lo phenotype of the population of interest is identified, why not use it for the remainder of the studies. Sometimes IFN γ is used, other times not. The same is true with AMREG . This inconsistency complicates interpretation.
14. In line 248 the authors use the terms induced but this has not been demonstrated as opposed to survival/proliferation.
15. It is not clear why the number of PD-1hiIFN γ +FoxP3 cells increases when death is blocked if these cells are resistant to death. Doesn't this mean that a portion of these cells is also susceptible to death and preventing it allows them to survive and proliferate?

16. Which cells within the PD-1 subset are those that survive? The authors focus on IFN γ in 4C, but in the analysis of Ki-67, all of the PD-1^{hi}FoxP3⁺ cells. The IFN γ producers are a small subset of that population.

17. Line 300 states IL-33 and IL-6 not detected, although data are presented in S11. Why was Efa added in these experiments when the TLR ligands were included? The rationale for the experimental design should be included.

18. The authors conclude that PD-1 signaling increases stability of FOXP3 based on the data in Fig 6C, although these data seem to show a difference in the absolute expression of FoxP3, i.e. the percent positive. Increased stability would be expected to increase the MFI. Are there data or previous reports that FoxP3 is rapidly turned over in negative populations that then with stabilization becomes positive, i.e. turnover controls positive versus negative?

Minor points

- It would be helpful to state the activation conditions used in the legend. Also the phenotype on which Tregs are sorted using the kit should be stated in the methods.
- “The” missing line 134 before inflammasome.
- Panels appear to be switched in FS3.
- Need to reference figure 2B line 147
- There may be an error in legend of S6B. Is CD25 in the gating strategy? It is not mentioned in the text or stated as a pre-gate in the legend.
- The colors in the legend and graph do not match in S12.

Reviewer #4 (Remarks to the Author):

In this manuscript, Bhaskaran et al., are seeking to understand the mechanisms of the residual systemic inflammation and mucosal immune dysfunction that are observed in people living with HIV, even in those in which the virus is suppressed with ART. They report that an altered immune landscape involving upregulation of TLR and inflammasome signaling, local CD4⁺ T cell hyperactivation, and counterintuitively, an enrichment of CD4⁺CD25⁺FOXP3⁺ regulatory T cells (Tregs) can be observed in the oral mucosa of HIV⁺ patients on ART. They are using human oral tonsil cultures to show that HIV infection causes an increase in a unique population of FOXP3⁺ cells expressing PD-1, IFN- γ , Amphiregulin (AREG), and IL-10. They also document persistence of these cells even in the presence of ARVs and they show that they underwent further expansion driven by TLR-2 ligands and IL-1 β . IL-1 β also promoted PD-1 upregulation in AKT1 dependent manner. PD-1 stabilized FOXP3 and AREG expression in these cells through a mechanism requiring the activation of Asparaginyl Endopeptidase (AEP). Importantly, these FOXP3⁺ cells were incapable of suppressing CD4⁺ T cells in vitro. Further, they report that higher levels of PD-1, IFN- γ , Amphiregulin (AREG), and IL-10 expressing FOXP3⁺ cells occur in HIV-infected subjects and they strongly correlate with CD4⁺ T cell hyperactivation, suggesting an absence of CD4⁺ T cell regulation in oral mucosa. As such, the manuscript addresses issues related to FOXP3⁺ cell dysregulation and identifies a link in the positive feedback loop of oral mucosal immune activation in HIV⁺ patients on ART.

The manuscript is interesting, well crafted and of potential importance to the field.

There are several issues with the data that should be addressed.

1. One potential major issue is that, as shown in figure 3, they first define Tregs as CD25⁺ CD4⁺ T cells instead of cd25^{high}, but also they define Tregs based on cd25 cd127 and then they assess their fox p3 expression. And tregs should be foxp3 pos by definition. This strategy should be revised or at least discussed.
2. The authors should assess the internalization of IL-10R and modulation of downstream targets, which was published to happen in the intestinal mucosa in SIV infected animals by Pahar, Veazey

and Lackner doi: 10.1128/JVI.01757-14

3. Is it possible that conversion of CD4 to pTreg phenotype could add to the change in frequencies? AKT activation is linked to pTreg differentiation.
4. They should discuss the role of AKT/FOXO3 and IFN- γ in Treg suppressive pathway and potentially look towards multiple sclerosis where they've shown that constitutively active AKT leads to increased IFN production instead of decreasing it and blocking the pathway suppresses IFN and recovers suppressive abilities of Tregs. doi.org/10.15252/embr.201541905 This can also be an experiment. They've already done a suppression assay, they should do it again, but blockade the pathway or IFN- γ secretion and see if it fixes the suppressive functionality
5. Further, they should discuss the inherent survival advantage that Tregs have over CD4 during HIV infection due to their difference in apoptotic gene expression
6. They should proof the paper again to reduce the incidence of punctuation errors (primarily spacing).

Rebuttal response:

We thank all the reviewers for their time and effort in reading our manuscript carefully and providing their constructive comments. We have provided new evidence from additional experiments addressing their concerns and supporting the findings of our study. We have made revisions in the manuscript and the textual changes are highlighted in gray color. We respond to their comments in a point-by-point fashion below:

REVIEWER COMMENTS

Reviewer #2 (Remarks to the Author):

In the manuscript, “PD-1 dependent expansion of Amphiregulin+FoxP3+ cells is associated with oral immune dysfunction in HIV patients on therapy” by Bhaskaran et al., the authors characterized a novel CD4 T-cell population they exclusively find the tonsils of HIV patients under anti-viral therapy. This T-cell population shows all the hallmarks of regulatory T-cells (Tregs) as these cells express high levels of CD25 and FoxP3 combined with a low expression of CD127. In addition, this T-cell population expresses the activation marker PD-1 and the cytokines IFN γ , IL-10 and Amphiregulin. Strikingly, the frequency of this T-cell population directly correlates with a general hyperactivation state (based on CD38 and MHC-II expression) of effector CD4 T-cells. The experiments are all performed in a technical sound way and the presented data appear quite intriguing.

Unfortunately, all presented data remain purely descriptive in kind. Thus, it remains doubtful whether the conclusions of the manuscript are justified. For instance, it is not clear whether an inflammatory environment causes the phenotype of these cells or whether the presence of these cells causes an inflammatory environment. That inflammatory cytokines lead to a positive feedback loop via the suppression of Treg function is well established and not specific for HIV. For instance, TLR-2 mediated Treg activation induces Treg proliferation and diminishes their suppressive capacity (Sutmuller et al. 2006), as the authors demonstrate in their manuscript.

Response: *The question of whether the inflammatory environment triggers Foxp3⁺ cells to become dysfunctional or whether dysfunctionality in T_{regs} precedes inflammation is without a doubt important to address. Based on our results, T_{reg} dysfunctionality is initially triggered by HIV- infection dependent IL-1 β cytokine up-regulation that induces proliferation of cell- death resistant PD-1^{hi} FOXP3⁺ cells. We have identified these dysfunctional T_{regs} to express other markers such as IFN- γ and AREG. Once FOXP3⁺ cells are dysfunctional, the immune system, having lost an immune-regulation arm, CD4⁺ T cells get hyperactivated and might exacerbate the inflammation further. While we and others have shown the role of TLR-2 in T_{reg} expansion and dysfunction, here we show that TLR-2 signaling alone doesn't lead to this phenotype. TLR-2 ligands only lead to further accumulation of PD-1^{high} cells and AREG upregulation in these cells in the context of HIV infection. These dysfunctional T_{reg} cells can undergo proliferation, despite ART and are found enriched in oral mucosa of HIV+ patients (Fig.8). The results shown in Fig.5A; see Uninfected +LPS, Uninfected +PG-LPS, Uninfected + HKGT, and Fig.R1 (new result) demonstrate that PD-1 and AREG expression require a concerted effort of HIV and TLR-2 signaling. So, to our*

knowledge, these findings are distinct from previous studies. Corroborating our in-vitro mechanistic data on dysfunctional T_{regs} , we also show for the first time that the oral mucosa of HIV+ individuals shows enrichment of TLR-2 signaling, inflammasome signaling, $PD-1^{high}T_{regs}$ with dysfunctional phenotype, AREG increases in saliva correlating with CD4 hyperactivation ex vivo (Fig.8). Together, these results showing the HIV-mediated T_{reg} dysfunction mechanism are novel and are important in understanding the oral mucosal dysregulation in HIV+ patients.

Thus, in order to verify their conclusions (Line 445 – 446), it might be important that the authors isolate “naïve”/CD62L expressing Tregs and determine whether indeed TLR-2 ligand mediated activation of Tregs combined with IL-1 induces their differentiation into Amphiregulin expressing Tregs. If so, then this could be reapplied to the situation in HIV patients.

Response: About 82-92% of $Foxp3^{+}$ cells in tonsils are of $CD45RO^{+}CD62L^{low}$ phenotype (Fig. R1A). As expected, $CD45RO^{neg}$ naïve T_{regs} were $CD62L^{high}$ and also $PD-1$ negative (Fig.R1A). As the reviewer suggested, to address whether $PD-1^{high}$ dysfunctional T_{regs} can be induced from naïve T_{regs} , we isolated $CD45RO^{neg}$ naïve T_{reg} cells from tonsils and infected them with HIV infection in the presence of TLR-2 ligand. The frequency of $PD-1^{high}$ cells and AREG expression are much lower in these cultures when compared to non-purified CD4 T cell cultures (Compare Fig.R1B with Fig.5A). Furthermore, the data in Fig.7 show that naïve T_{reg} cultures harbored significantly lower proportions of $PD-1^{+}IFN-\gamma^{+}$ cells (Fig.7A, B). Note that these $PD-1^{+}$ cells were $PD-1^{low}$ and not $PD-1^{high}$. Most importantly, these $PD-1^{low}FOXP3^{+}$ cells retained suppressive activity (Fig.7C, D), whereas $PD-1^{hi}IFN-\gamma^{+}T_{regs}$ that accumulated in HIV-infected cultures are dysfunctional.

Reduced $PD-1^{hi}IFN-\gamma^{+}AREG^{+}$ cell induction in naïve T_{reg} cultures may be attributed to the reduced ability of naïve T_{reg} cells to expand when compared to the pre-existing $CD45RO^{+}T_{regs}$ in mixed cultures. This notion is supported by previous literature (Booth et.al., *J.Immunol*, 2010, 184: 4317–4326, Different Proliferative Potential and Migratory Characteristics of Human CD4+ Regulatory T Cells That Express either CD45RA or CD45RO; DOI: 10.4049/jimmunol.0903781). Also, pre-existing $CD45RO^{+}T_{reg}$ cells may intrinsically have a higher potential to resist cell-death due to increased BCL-2 expression compared to $CD45RO^{neg}FOXP3^{+}$ cells (Fig. R1C). These data are included in Fig. S13 in the revised version of the manuscript. These $PD-1^{high}T_{reg}$ cells having a survival advantage (higher Bcl-2 expression- See Fig.3E, and Fig. R3) undergo further proliferation upon exposure to TLR-2 ligands and may contribute to further accumulation of dysfunctional T_{regs} . Therefore, it is conceivable that in mucosal tissues and tonsils that are known to be enriched with pre-existing $CD45RO^{+}FOXP3^{+}$ cells, these cells may contribute better to dysfunctional T_{reg} cells. However, the local induction from $CD45RO^{neg}FOXP3^{+}$ cells and recently activated $FOXP3^{+}$ cells cannot be ruled out. Taken together, these data show that while pre-existing $CD45RO^{+}FOXP3^{+}$ cells due to inherent survival advantage and enhanced proliferation might contribute more to the accumulation of dysfunctional T_{regs} , naïve $CD45RO^{neg}FOXP3^{+}$ cells can also be induced to become $PD-1^{high}$ cells.

Fig.R1
Fig.R1. PD-1^{hi}IFN- γ ⁺ FOXP3⁺ cell induction from naïve T_{regs} is enhanced by TLR-2 ligands in the context of HIV infection. A, C) Ex vivo flow cytometry analysis of tonsils (gated on CD4⁺FOXP3⁺ cells). B) CD4⁺CD45RO^{neg}CD127^{low}CD25⁺T_{reg} cells were sorted using three-step sorting of the tonsil cells. First, EasySepTM Human CD4⁺ T Cell Isolation Kit (STEMCELL Technologies) was used to isolate untouched CD4⁺ T cells using depletion of non-CD4 T cells. Then human CD45RO⁺CD4⁺ kit (Miltenyi biotech) was used to remove CD4 memory cells. Purified CD45RO^{neg} naïve (92%) were used for further purification of T_{regs} using human CD4⁺CD127^{low}CD25⁺ regulatory T cell kit (STEMCELL Technologies) (> 85% FOXP3⁺). These cells were activated with TCR stimulation and allowed to expand in the presence of TGF- β 1 (10 ng/ml) and IL-2 (100 U/ml). Some cells were infected with HIV on day 2 after TCR stimulation. Indicated cytokines or reagents were also added during this time. Flow cytometry was performed on day 7 after infection.

Minor comments:

- To measure the suppressive capacity of Tregs after an extended in vitro culture (four days) and HIV infection appears disingenuous as such culture conditions in no way reflect the phenotype of these cells directly ex vivo.

Response: *Suppressive capacity of T_{regs} is not always lost after an extended in vitro culture. For example, the control naïve CD127^{low}CD25⁺T_{regs} that retain their suppressive capacity (Fig.7C, D) were also stimulated and infected in the same manner. These are consistent with the previous literature (Pandiyana et al., 2007, Nature Immunology, PMID: 17982458 DOI: 10.1038/ni1536, Thornton 2000, J. Immunol, PMID: 10605010 DOI: 10.4049/jimmunol.164.1.183). Although it is conceivable that the phenotype of these cells may not resemble those isolated directly ex vivo, or those found in vivo, this was a proof-of-principle experiment that was performed to examine the functional capacity of PD-1^{high} T_{regs} after HIV infection. This was the only approach for the suppressive assay considering the limitations in the human cell/HIV infection system.*

We have also revised the Fig.7 legend better explaining how the control T_{regs} were obtained.

- Amphiregulin and T2 (the IL-33R) expression is typically associated with tissue resident Tregs. This aspect should be discussed in the manuscript.

Response: *The reviewer is right. These PD-1^{hi} T_{regs} responding to IL-1 β and expressing AREG resemble tissue T_{regs} induced by IL-33, another IL-1 superfamily cytokine. Similar to tissue T_{regs}, they may have*

non-suppressive roles and may function towards mucosal tissue repair during inflammation. We have included this discussion in the revised version of the manuscript. See line 424.

- PMA / Ionomycin is not a physiological stimulus for Tregs. Are these Tregs also producing pro-inflammatory cytokines, such as IL-1 and IFN γ , following TCR mediated activation, as the authors performed in Figure 3.

Response: We performed α -CD3/ α -CD28 re-stimulation controls, as a part of preliminary experiments while establishing the tonsil HIV cultures. Although not as robustly as PMA/Iono re-stimulated cells (as in Fig.3), CD4⁺ cells are capable of expressing IL-1 β , AREG, and IFN- γ even when re-stimulated with α -CD3/ α -CD28 (Fig. R2).

Fig.R2. α -CD3 and α -CD28 re-stimulation. CD4⁺ cells were activated with TCR stimulation and allowed to expand in the presence of TGF- β 1 (10 ng/ml) and IL-2 (100 U/ml). Some cells were infected with HIV on day 2 after TCR stimulation. Indicated cytokines or reagents were also added during this time. Cells were re-stimulated with α -CD3 and α -CD28 (5 μ g/ml each) on day 7 after infection, before the cells were collected for flow cytometry and supernatants were collected for ELISA(A,B,C). PD-1^{high} or PD-1 negative cells were gated during analysis of the HIV infected cells in (C).

- Figure 3a – how do you get FoxP3 negative Tregs..?

Response: These are gated on FOXP3 negative fraction during analysis. As the reviewer can appreciate, not all the sorted CD25^{high} cells are FOXP3⁺. After sorting, only 85-90% of the cells were FOXP3⁺ in purified fraction. In figure 3A we have gated on Foxp3 negative cells in the analyses. We have revised the Fig.3 legend to explain the gating better.

Reviewer #3 (Remarks to the Author):

This manuscript by Bhaskaran et al identified the presence of a population of Tregs that PD-1, IFN γ , Amphiregulin and IL-10. These cells were notable in that they were incapable of suppressing CD4⁺ T cells in vitro. A major concern is the conclusion regarding induction of cells. This is not formally shown as populations are not sorted to definitively show the differentiation of one type of cell into another.

Response: As the reviewer suggested, to address whether PD-1^{high} dysfunctional T_{regs} can be induced from naïve T_{regs}, we isolated CD45RO^{neg} naïve T_{reg} cells from tonsils and infected them with HIV infection in the presence of TLR-2 ligand (See Fig.R1). These results show that dysfunctional PD-1⁺AREG⁺T_{regs} can also be induced from naïve T_{reg} cells during HIV infection. These PD-1^{high} T_{reg} cells having a survival

advantage (higher Bcl-2 expression- See Fig.3E, and Fig.R3) undergo further proliferation upon exposure to TLR-2 ligands and may contribute to further accumulation of dysfunctional T_{regs} . Taken together, these data show that while pre-existing $CD45RO^+FOXP3^+$ cells contribute more to the accumulation of dysfunctional T_{regs} , naïve $CD45RO^{neg}FOXP3^+$ cells can also be induced to become $PD-1^{high}$ and $AREG^{high}$ cells.

Additionally, we have also sorted $PD-1^+T_{reg}$ and $PD-1^{neg}T_{reg}$ cells from tonsils and examined the expression of secondary markers such as $IFN-\gamma$ and $AREG$ with and without infection (See Fig.R3). Consistent with our hypothesis and the results in Fig.3, purified $PD-1^+T_{reg}$ cells showed higher $IFN-\gamma$ and $AREG$ expression, compared to $PD-1^{neg}T_{reg}$ cells (Fig.R3A, B). $PD-1^+T_{reg}$ cells also showed higher and Ki-67, HIV-GFP and BCL-2 expression than $PD-1^{neg}T_{reg}$ cells (Fig. R3C, D, E). These data support the notion that $PD-1^+T_{regs}$ although has high infection susceptibility, may intrinsically survive and proliferate better with HIV infection, leading to the accumulation of dysfunctional T_{regs} . Interestingly, a small proportion $PD-1^{neg}T_{reg}$ population can also upregulate $PD-1$ and $IFN-\gamma$ in the context of HIV infection (but not TLR-2 stimulation alone). These induced cells also show higher proliferation with TLR-2 and $IL-1\beta$ stimulation in the context of HIV infection (Fig. R3C). It is conceivable that in mucosal tissues and tonsils that are known to be enriched with pre-existing $PD-1^+FOXP3^+$ cells, these cells may contribute better to the accrual of dysfunctional T_{reg} cells than naïve T cells. However, the local induction from $PD-1^{neg}FOXP3^+$ cells cannot be ruled out in the process. Taken together, these data show that while pre-existing $PD-1^+FOXP3^+$ cells contribute more to the accumulation of dysfunctional T_{regs} , $PD-1^{neg}FOXP3^+$ cells can also be induced to become $PD-1^{high}$ cells expressing high levels of $IFN-\gamma$ and $AREG$. These new data are included in **Fig. S14** in the revised version of the manuscript.

Fig.R3
Fig.R3. PD-1^{hi}IFN-γ⁺FOXP3⁺ cell induction and proliferation are enhanced by TLR-2 ligands in the context of HIV infection. CD4⁺PD-1^{neg}CD127^{low}CD25⁺ T_{reg} and CD4⁺PD-1⁺CD127^{low}CD25⁺ T_{reg} cells were sorted using three-step sorting of the tonsil cells. EasySep™ Human CD4⁺ T Cell Isolation Kit (STEMCELL Technologies) was used to isolate untouched CD4⁺ T cells using depletion of non-CD4 T cells. Anti-Biotin multisort kit (Miltenyi biotech) was used to positively sort anti-PD-1-biotin labelled cells. The beads were removed from positively sorted cells. Purified CD4⁺PD-1⁺ and CD4⁺PD-1^{neg} (~90-95%) were used for further purification of T_{regs} using human CD4⁺CD127^{low}CD25⁺ regulatory T cell kit (STEMCELL Technologies) (> 85% FOXP3⁺). These cells were activated with TCR stimulation and allowed to expand in the presence of TGF-β1 (10 ng/ml) and IL-2 (100 U/ml). Some cells were infected with HIV on day 2 after TCR stimulation. Indicated cytokines or reagents were also added during this time. Flow cytometry was performed on day 7 after infection to determine PD-1, IFN-γ, (A), AREG (B), Ki-67 (C), and GFP (D). (E) BCL-2 expression on day 0 (d0) and d7 after HIV infection.

Additionally, in several places there is a comparison with PBMC although the rationale for this and the significance is not clear beyond Fig. 1 and is a distraction from the main point of the study.

Response: We show data from PBMC in Fig. 1, 2D, and S5. In figures 1 and 2D, ex vivo PBMC cells served as controls for oral mucosa in which only the latter show the enrichment of FOXP3⁺ cells and cytokine dysregulation. Again, in Fig.S5, PBMC were used as controls while profiling the tonsil cell populations, such as TFH and TFR that we used in the experiments.

Finally, in the absence of the ability to demonstrate reduced suppressive activity in the cells from patients on cART, whether the in vitro results, while intriguing, are the case in HIV+ individuals is speculative. Based on the in vitro studies, it would seem that the model is that infection early on results in the presence of ineffective Tregs that is

maintained for long periods of time. Is there an indication that these cells could be long lived in a way that would fit this model? While challenging to prove, the authors should discuss this or clearly define the model if this is not the case.

The reviewer brings up an important point. Our results showing that dysfunctional T_{reg} cells with high expression of BCL-2 and KI-67, and resistance to apoptosis (Fig.3E,4D), suggest that these cells may be long-lived and may undergo continuous cycling. Moreover, we have also shown that PD-1^{high}FOXP3+ cells expressing IFN- γ and AREG are indeed enriched in HIV+cART patients correlating with CD4 hyperactivation (Fig.8), which would imply that these may be long-lived and dysfunctional in these patients. Supporting this notion, we here show new results from three HIV+ patients whose FOXP3+PD-1^{high} cells also express high levels of BCL-2 (Fig.R4), a pro-survival protein. As the reviewer suggested, we have discussed this model in the revised manuscript. See line 494.

Fig.R4

Fig.R4. PD-1^{hi} cells have increased BCL-2 in oral mucosa of HIV+cART patients *ex vivo*. HOILs from gingival mucosa from three HIV+ patients on cART were processed for flow cytometry. BCL-2 expression in CD3⁺CD4⁺FOXP3⁺ gated HOIL cells.

Also it is not clear if the act of infection or some other effect of the infection culture is responsible for the loss of suppressive activity (Fig. 7) as infected Tregs vs non-infected Tregs are not assessed. Finally, a number of the data shown appear to confirm those previously published and as presented the studies are at times difficult to follow. These issues significantly weaken the impact of the study.

Response: In Fig. 7, we examined the function of CD4⁺CD25⁺PD-1^{high}FOXP3⁺ cells that were induced during HIV infection *in vitro* and compared them with purified naïve CD4⁺CD25⁺CD127^{low}FOXP3⁺ cells activated the same length of time and infected the same manner. The data in Fig. 7 also show that these naïve T_{reg} cultures harbored significantly lower proportions of PD-1^{hi}IFN- γ + cells (Fig. 7A, B), but retained suppressive activity (Fig. 7C, D). Note that these cells have the highest infection susceptibility as well (Fig. 3A). These results demonstrate that infection *per se* does not affect their suppressive capacity. Moreover, the impaired suppressive ability is rather attributed to the high accumulation of CD4⁺CD25⁺PD-1⁺FOXP3⁺ cells. Early IL-1 β expression drives induction and proliferation of these dysfunctional cells in HIV-infected cultures. TLR-2 alone does NOT lead to the accumulation of these cells (Fig. 5A, see Uninfected). To our knowledge, the role of IL-1 β in HIV-mediated dysregulation of T_{regs} and oral mucosal dysfunction is novel and has not been shown elsewhere.

In addition the following are concerns are noted. 1. From the methods it appears that >75% of the HIV+ individuals in Fig.1 had/have infection/inflammatory conditions independent of HIV. Wouldn't this contribute to the increase inflammation? Wouldn't a similar set of non-HIV+ individuals be the appropriate comparator?

Response: This is an important question. As the reviewer pointed out, a majority of HIV+ patients had previous oral lesions. As we have noted in the manuscript, it is known that oral complications such as periodontitis are of increased incidence and severity in HIV+ individuals even after suppressive HIV therapy. One rationale of the study was to determine the underlying oral mucosal immune cell alterations because it not clear now how HIV+ status by itself contributes to the increased prevalence of oral inflammatory diseases. Before we embarked on this study, we had profiled FOXP3+ cells in gingival mucosa from both chronic and acute periodontitis non-HIV patients comparing them with healthy individuals. Although we found increases in Th17 cells in periodontitis non-HIV patients, as shown previously (Dutzan 2016, Characterization of the human immune cell network at the gingival barrier, Mucosal Immunol 9(5): 1163–1172. doi:10.1038/mi.2015.136), there were no significant changes in the frequency of FOXP3+ cells in their gingiva (Fig. R5). These data are presented here and are included in Fig.S4. These results show that previous inflammation does not by itself correlate or contribute to FOXP3+ T_{reg} cell increase and their potential dysfunction. From these results, we can infer that FOXP3+ T_{reg} dysregulation that we observe in HIV+ patients is not a result of other inflammatory conditions.

Fig.R5

Fig.R5. No T_{reg} alterations were observed in the oral mucosa of periodontitis patients *ex vivo*. HOILs from gingival mucosa of healthy control (healthy; n=8) or periodontitis patients (Perio. n= 9) were processed for flow cytometry. CD25 and FOXP3 expression in CD3⁺CD4⁺ gated HOIL cells (A). Statistical analyses comparing % CD4, Th17 cells and T_{regs}, comparing the two groups. * P< 0.05; Mann Whitney test.

2. There is some question as to the gating for positivity. At time it looks as if the gate runs through the middle of a population (CD38 in 1E) or includes negative cells (FoxP3 in Fig 4A).

Response: The gates were assigned based on the unstained controls, FMO controls, and non-restimulated control cells, for each experiment. Because tissue cells (oral mucosa) may have some autofluorescence, we also performed PBMC staining control cells in parallel. We chose the most appropriate gating based on an objective evaluation using these controls when establishing the staining protocol for experiments. However, the reviewer is right about the gate running through the middle of the CD38⁺ population. We have moved it to the best possible extent as the reviewer suggested and revised Fig.2D. Here we also show the un-stain, FMO controls for the CD38 and HLA-DR staining shown in

Fig.2D, which rationalize the position of the gate (Fig.R6A). We also performed the staining in PBMC (Fig.R6A, PBMC controls; last column) based on which we have moved the gate very high. Note that the gate doesn't run in the middle of any population in PBMC. We have included the gating strategy in methods and legends (Fig.S19).

Fig.R6. Staining controls. HOIL were processed *ex vivo* for flow cytometry. Gating strategy using unstain, FMO controls, or PBMC control cell staining for HLADR and CD38 staining (A), and Non-restimulated cell control for IFN- γ and PD-1 staining (B), are shown.

In Fig S8 why is the cut-off for IFNgamma so high if it based on the FMO control? Perhaps a better approach would be the non-restimulated cells which take into account the non-specific staining of the antibody. Also the authors should use caution when it appears the overall background of a sample is higher than another as is the case for 2D (apparent MFI of the double negative population tope panel) as it may skew the results.

Response: With relevance to IFN- γ staining in S8, we used a higher gate because we found some shift in non-restimulated cells, exactly as the reviewer suggested(Fig.R6B). We have now included this control also in Fig.S8 in the revised version of the manuscript.

3.The two very distinct populations present in the HIV+cART CD38+DR+ gate merits comment.

Response: We found distinct populations only in oral mucosa and not in PBMC (See Fig.R6A). It may be related to the downregulation of HLA-DR (MHC) in CD4⁺ T cells. While HLA-DR downregulation in monocytes is associated with immune-suppression, it remains to be seen if this is the case for CD4⁺ T cells. This speculation is now included in the discussion section. See line 471.

4.It is not clear why the authors state IL-6 and IL-1 are too low to detect when they show data for these measurements in S3.

Response: This was true for IL-6 in saliva where values in most of the samples were less than 5 pg/ml, lower than the sensitivity of the ELISA. This could very well reflect the actual absence of this cytokine in their saliva. Therefore, we will remove this statement in the revised version of the manuscript.

5. The Suppl. Figures would benefit from expanded legends that include the experimental detail as opposed to referring back to other figures, including the number tested.

Response: We have included the experimental details in all figure legends for the supplementary figures in the revised version of the manuscript.

6. The CD14 data in 2B are not described in the results section.

Response: We apologize for this oversight. We have corrected this and included this result in the revised version of the manuscript.

7. The authors conclude there are alterations in TLR-2 signaling (Line 149), although this is not directly shown.

Response: Fig. 1B and 1D transcriptomics data show enrichment of TLR-2 mRNA and TLR-2 signaling genes in oral mucosa of HIV+ patients. Also, increased s-TLR-2 ligands we observed in HIV+ patients (Fig. 2B) may be due to TLR-2 shedding as a consequence of increased pro-inflammatory signaling downstream to TLR-2 signaling (Langjahr et. al., 2014, Metalloproteinase-dependent TLR2 ectodomain shedding is involved in soluble toll-like receptor 2 (sTLR2) production. PLoS One. 22;9(12):e104624. DOI: 10.1371). We have now included this sentence in the discussion. See line 477.

8. Statistical analysis using multiple samples would be required to make statements of significance in lines 192-95.

Response: We have included the requested statistical analysis in **Fig. S5E** in the revised version of the manuscript.

9. The conclusion that memory Tregs are the most permissible would seem to require sorting and comparison of this population and CD45RO negative cells.

Response: We sorted these cells as suggested (as in Fig. R1) and compared their infection permissibility. The results show that CD45RO⁺ T_{reg} cells are moderately higher than CD45RO^{neg} T_{reg} cells in their infection susceptibility (Fig. R7). Also, a fraction of CD45RO^{neg} T_{reg} cells upregulated CD45RO. Therefore, we have removed this claim in the revised version of the manuscript.

Fig.R7. CD45RO⁺FOXP3⁺ cells are moderately more permissive to HIV infection than naïve T_{regs}. EasySep™ Human CD4⁺ T Cell Isolation Kit (STEMCELL Technologies) was used to isolate untouched CD4⁺ T cells using depletion of non-CD4 T cells. Anti-Biotin multisort kit (Miltenyi biotech) was used to positively sort anti-CD45RO-biotin labelled cells. The beads were removed from positively sorted cells. Purified CD4⁺CD45RO⁺ and CD4⁺CD45RO^{neg} (~90-95%) were used for further purification of T_{regs} using human CD4⁺CD127^{low}CD25⁺ regulatory T cell kit (STEMCELL Technologies) (> 85% FOXP3⁺). These cells were activated with TCR stimulation and allowed to expand in the presence of TGF-β1 (10 ng/ml) and IL-2 (100 U/ml). Some cells were infected with HIV on day 2 after TCR stimulation. Indicated cytokines or reagents were also added during this time. Flow cytometry was performed to determine GFP-expression in CD4⁺FOXP3⁺ cells 6 days post-infection.

10. It is not clear what led to the statement “They also suggest that HIV infection may alter the functionality of FOXP3⁺ cells in oral mucosal lymphoid compartments.” as no functional data have been shown. line 215

Response: In Fig. 7, we have demonstrated that CD4⁺CD25⁺PD-1^{high}FOXP3⁺ cells that were induced and expand during HIV infection in vitro have impaired suppressive function, demonstrating that PD-1^{high}FOXP3⁺ cells are dysfunctional. TLR-2 alone does NOT lead to the accumulation of these cells (Fig. 5A, see Uninfected). To our knowledge, the role of IL-1β in HIV mediated dysregulation of T_{regs} and oral mucosal dysfunction is novel and has not been shown elsewhere. The data showing the accumulation of PD-1^{high} T_{reg} cells (Fig. 8A, B, D) correlating with CD4 hyperactivation (Fig. 1, 8J, 8K) despite increased T_{regs} (Fig 2) in oral mucosa of HIV⁺ patients, and the accumulation of dysfunctional T_{reg} cells during HIV infection in vitro, when taken together, led us to suggest that HIV infection may alter the functionality of FOXP3⁺ cells in oral mucosal lymphoid compartments. However, we have removed that speculative statement from the “Results” section.

Fig.R8

Fig.R8. PD-1^{high} FOXP3⁺ cell population harbor higher frequency of IFN- γ + cells in HIV-infected cultures . Purified CD4⁺ cells were activated with TCR stimulation and allowed to expand in the presence of TGF- β 1 (10 ng/ml) and IL-2 (100 U/ml). They were infected with HIV on day 2 after TCR stimulation. Flow cytometry was performed to determine IFN- γ in CD4⁺FOXP3⁺ cells, gated on PD-1 -high, -intermediate and -low populations, 6 days post-infection. Contour plot data from two independent tonsil donors (A). Statistical analysis from 4 representative donors (n=4) (B). * P< 0.05.

11.The very low level of IFN γ in the FoxP3⁺ cells merits noting in the results.

Response: We also noted that Foxp3⁺PD-1high cells appeared to show IFN- γ (MFI) in many experiments. However, this was not observed consistently in all the experiments using tonsils from different donors. Here, we have summarized the data from 4 independent experiments/donors (Fig.R8). We have now included this in the discussion. See line 428.

12.It was surprising that the authors did not sort PD-1 high versus low TFR and infect the cultures to address the interrelationship between these two populations with regard to infection and the bcl-2 expression and susceptibility to death prior to and after infection. This would provide definitive data regarding the PD-1high population.

Response: We agree with the reviewer. As they suggested, here we have sorted PD-1⁺T_{reg} and PD-1^{neg}T_{reg} cells from tonsils and examined the expression of secondary markers such as IFN- γ and AREG, with and without infection (See. Fig. R3). Consistent with our hypothesis and the data in Fig.3, purified PD-1⁺T_{reg} cells showed higher IFN- γ and AREG expression, compared to PD-1^{neg}T_{reg} cells (Fig. R3A, B). PD-1⁺T_{reg} cells also showed higher Ki-67, HIV-GFP, and BCL-2 expression than PD-1^{neg}T_{reg} cells(Fig. R3D, E). These data support the notion that PD-1⁺T_{regs} although has high infection susceptibility, may intrinsically survive and proliferate better with HIV infection, leading to the accumulation of dysfunctional T_{regs}. Interestingly a small proportion of PD-1^{neg}T_{reg} cells can also upregulate PD-1 and IFN- γ in the context of HIV infection (and not TLR-2 stimulation alone). They also show higher proliferation with TLR-2 and IL-1 β stimulation in the context of HIV infection, but not as much as purified PD-1⁺T_{regs} (Fig.R3C). It is conceivable that in mucosal tissues and tonsils that are known to be enriched with pre-existing PD-1⁺FOXP3⁺ cells, these cells may contribute better to the accrual of dysfunctional T_{reg} cells than naïve T_{reg}

cells. However, the local induction from PD-1^{neg} FOXP3⁺ cells cannot be ruled out in the process. Taken together, these data show that while pre-existing PD-1⁺ FOXP3⁺ cells contribute more to the accumulation of dysfunctional T_{regs}, PD-1^{neg} FOXP3⁺ cells can also be induced to become PD-1^{high} cells expressing higher levels of IFN- γ and AREG and become dysfunctional.

13. Once the PD-1^{hi}, CD25, CXCR5, AMREG, CD127^{lo} phenotype of the population of interest is identified, why not use it for the remainder of the studies. Sometimes IFN γ is used, other times not. The same is true with AMREG. This inconsistency complicates interpretation.

Response: Almost >35-68% of the PD-1^{hi}FOXP3⁺ cells express IFN- γ while the PD-1^{neg} population has lower levels of IFN- γ producing cells (~12-20%). See Fig. 3D, or 4C for examples (see also Fig.R8, Fig R3). As long as they are FOXP3⁺, whether we include IFN- γ ⁺ population or not, the differences with PD-1^{high} vs low cells were consistent in terms of other proteins they express, e.g. KI67, BCL-2, etc. Therefore, we used the “PD-1^{high}” marker alone for the experiments starting from Fig.6. In T_{reg} sort experiments, PD-1^{high}CD25⁺ cells purified from HIV infected cultures were 80-88% FOXP3⁺, IFN- γ ⁺ (52%) (Fig.S18), AREG^{high}, and were used as PD-1^{high} CD25⁺FOXP3⁺ cells (mentioned in methods). Nevertheless, we have revised the manuscript to consistently refer to them as “PD-1^{hi} IFN- γ ⁺ AREG^{high} FOXP3⁺ cells” in our results and discussion.

14. In line 248 the authors use the terms induced but this has not been demonstrated as opposed to survival/proliferation.

Response: A small population of CD45RO^{neg} FOXP3⁺ cells and PD-1^{neg} FOXP3⁺ cells can become PD-1^{high}IFN- γ ⁺AREG^{high} cells, (See Fig.R1 and R3). It is conceivable that in mucosal tissues and tonsils that are known to be enriched with pre-existing CD45RO⁺FOXP3⁺ cells, survival, and proliferation of these cells during HIV infection might contribute better to the dysfunctional PD-1^{high}IFN- γ ⁺AREG^{high} FOXP3⁺T_{reg} cells than the induction process. However, we cannot exclude the “induction” of T_{regs} also contributing to PD-1^{high}IFN- γ ⁺AREG^{high} FOXP3⁺ cell accumulation in mucosal tissues. Nevertheless, in the above-referred sentence, we have changed “induction” to “accumulation”.

15. It is not clear why the number of PD-1^{hi}IFN γ +FoxP3 cells increases when death is blocked if these cells are resistant to death. Doesn't this mean that a portion of these cells is also susceptible to death and preventing it allows them to survive and proliferate?

Response: Yes, the reviewer is right. PD-1^{high}IFN- γ ⁺AREG^{high}FOXP3⁺ T_{reg} cells are more cell-death resistant compared to PD-1^{neg} cells, but a proportion of them (36%) still undergo cell death (Fig 3E).

Moreover, a further increase in the absolute number of cells is contributed by proliferation. See the % of Ki-67 compared to PD-1^{low} cells. (Fig.4D). We have now provided new evidence using sorted PD-1^{high} vs PD-1^{low} T_{reg} cells, showing higher BCL-2 and Ki-67 expression in purified PD-1^{high} T_{regs}, which further confirms these findings (See Fig.R3).

16. Which cells within the PD-1 subset are those that survive? The authors focus on IFN γ in 4C, but in the analysis of Ki-67, all of the PD-1^{hi}FOXP3⁺ cells. The IFN γ producers are a small subset of that population.

Response: Almost >35-68% of the PD-1^{hi}FOXP3⁺ cells express IFN- γ . See Fig.3D, 4C, S18 for examples (see also Fig.R8, Fig R3). Only the PD-1^{neg} population has lower levels of IFN- γ producing cells (~12-20%). As long as they are FOXP3⁺, whether we include IFN- γ or not, the differences with PD-1^{high} vs PD-1^{neg} cells were consistent in terms of other proteins they express, e.g. AREG, KI67, BCL-2, etc (See also the new evidence in Fig.R3). Therefore, we used the "PD-1^{high}" marker alone for the experiments starting from Fig.6.

While our data show that PD-1^{high}FOXP3⁺ T_{reg} (that also express high levels of IFN- γ , AREG, BCL-2, and Ki-67) are more cell-death resistant compared to PD-1^{neg} cells, a proportion of them (36%) still undergo cell death (Fig 3E). We currently do not have conclusive evidence to show which sub-population of PD-1^{hi}FOXP3⁺ cells survive better. We will address this in our future experiments.

17. Line 300 states IL-33 and IL-6 not detected, although data are presented in S11.

Response: We apologize for the typo. We meant to say that there were no changes in IL-33 and IL-6 by HIV infection. We have changed this in the revised manuscript. See line 251.

Why was Efa added in these experiments when the TLR ligands were included? The rationale for the experimental design should be included.

Response: Because the rationale of the study was to determine the underlying oral mucosal dysregulation in HIV+cART patients, we wanted to mimic the effect of HIV in the presence and absence of HIV inhibitor (Efa). Note that our study participants in the HIV+ cohort are ART-treated, so their cells will exhibit features in the context of the suppressed virus. Considering the different anti-viral regimens that patients take, our in vitro experiments may not be exactly physiologically relevant, but may show similarity to HIV+cART patient samples. Nevertheless, it is the best approach with the available options of the human cell/HIV infection system. We have included this rationale in the revised manuscript. See line 413.

Also, from a technical point of view, in some experiments, HIV infection caused high levels of cell-death limiting further analysis of multi-parametric flow cytometry analysis of enough viable cells. We added Efa

to prevent HIV-induced cell-death so that we can have enough cells for analysis of these proteins. Considering that we already established the function of HIV alone in Fig-4, we performed the subsequent experiments with Efa.

18. The authors conclude that PD-1 signaling increases stability of FOXP3 based on the data in Fig 6C, although these data seem to show a difference in the absolute expression of FoxP3, i.e. the percent positive. Increased stability would be expected to increase the MFI. Are there data or previous reports that FoxP3 is rapidly turned over in negative populations that then with stabilization becomes positive, i.e. turnover controls positive versus negative?

Response: FOXP3 instability has been shown to cause complete loss of the protein as well as lowering of expression (Control of Foxp3 stability through modulation of TET activity. Xiaojing Yue, Sara Trifari, Tarmo Äijö, Ageliki Tsagaratou, William A. Pastor, Jorge A. Zepeda-Martínez, Chan-Wang J. Lio, Xiang Li, Yun Huang, Pandurangan Vijayanand, Harri Lähdesmäki, and Anjana Rao). In Fig. 6C, with PD-L1Fc and AEP inhibitor, we not only observed an increase in % of FOXP3+ cells but also observed an increase in MFI among FOXP3+ cells. We have now added the Geo Mean-MFI data in Fig. 6C. Furthermore, Fig. 6A shows data on gated FOXP3+ cells, demonstrating further that PD-1^{high} cells also have increased FOXP3 MFI (see 3rd panel).

We now provide additional evidence to show that purified PD-1^{neg} cells that were activated and infected as in Fig.R3, lose FOXP3, which further confirms that PD-1 is required for Foxp3 retention (Fig.R9). Taken together, our data support the notion that PD-1 increases FOXP3 stability by enhancing the FOXP3 expression levels in FOXP3⁺ cells and retaining the positivity of FOXP3 expression. These data are included in Fig.S17 in the revised version of the manuscript.

Fig.R9

Fig.R9. PD-1^{neg} FOXP3⁺ cells lose FOXP3 expression in the context of HIV infection. CD4⁺PD-1^{neg}CD127^{low}CD25⁺ T_{reg} and CD4⁺PD-1⁺CD127^{low}CD25⁺ T_{reg} cells were sorted using three-step sorting of the tonsil cells. EasySep™ Human CD4+ T Cell Isolation Kit (STEMCELL Technologies) was used to isolate untouched CD4⁺ T cells using depletion of non-CD4 T cells. Anti-Biotin multisort kit (Miltenyi biotech) was used to positively sort anti-PD-1-biotin labelled cells. The beads were removed from positively sorted cells. Purified CD4⁺PD-1⁺ and

CD4⁺PD-1^{neg} (~90-95%) were used for further purification of T_{regs} using human CD4⁺CD127^{low}CD25⁺ regulatory T cell kit (STEMCELL Technologies) (> 85% FOXP3⁺). These cells were activated with TCR stimulation and allowed to expand in the presence of TGF-β1 (10 ng/ml) and IL-2 (100 U/ml). Some cells were infected with HIV on day 2 after TCR stimulation. Flow cytometry was performed to determine FOXP3 expression on day 0 (d0) and d7 after HIV infection.

Minor points

-It would be helpful to state the activation conditions used in the legend. Also the phenotype on which Tregs are sorted using the kit should be stated in the methods.

Response: These are now mentioned in legends and methods in the revised version of the manuscript.

-“The” missing line 134 before inflammasome.

Response: We have corrected this in revised version of the manuscript. See line 111.

-Panels appear to be switched in FS3.

Response: We have corrected this in revised version of the manuscript.

-Need to reference figure 2B line 147

Response: We have corrected this in revised version of the manuscript. See line 479.

-There may be an error in legend of S6B. Is CD25 in the gating strategy? It is not mentioned in the text or stated as a pre-gate in the legend.

Response: We apologize for the typo. We have corrected this to CD45RO in the revised version of the manuscript.

-The colors in the legend and graph do not match in S12.

Response: We have corrected this in the revised version of the manuscript.

Reviewer #4 (Remarks to the Author):

In this manuscript, Bhaskaran et al., are seeking to understand the mechanisms of the residual systemic inflammation and mucosal immune dysfunction that are observed in people living with HIV, even in those in which the virus is suppressed with ART. They report that an altered immune landscape involving upregulation of TLR and inflammasome signaling, local CD4⁺ T cell hyperactivation, and counterintuitively, an enrichment of CD4⁺CD25⁺FOXP3⁺ regulatory T cells (Tregs) can be observed in the oral mucosa of HIV⁺ patients on ART.

They are using human oral tonsil cultures to show that HIV infection causes an increase in a unique population of FOXP3+ cells expressing PD-1, IFN- γ , Amphiregulin (AREG), and IL-10. They also document persistence of these cells even in the presence of ARVs and they show that they underwent further expansion driven by TLR-2 ligands and IL-1 β . IL-1 β also promoted PD-1 upregulation in AKT1 dependent manner. PD-1 stabilized FOXP3 and AREG expression in these cells through a mechanism requiring the activation of Asparaginyl Endopeptidase (AEP). Importantly, these FOXP3+ cells were incapable of suppressing CD4+ T cells in vitro. Further, they report that higher levels of PD-1, IFN- γ , Amphiregulin (AREG), and IL-10 expressing FOXP3+ cells occur in HIV-infected subjects and they strongly correlate with CD4+ T cell hyperactivation, suggesting an absence of CD4+ T cell regulation in oral mucosa. As such, the manuscript address issues related to FOXP3+ cell dysregulation and identifies a link in the positive feedback loop of oral mucosal immune activation in HIV+ patients on ART. The manuscript is interesting, well crafted and of potential importance to the field.

There are several issues with the data that should be addressed.

1. One potential major issue is that, as shown in figure 3, they first define Tregs as CD25+ CD4+ T cells instead of cd25^{high}, but also they define Tregs based on cd25 cd127 and then they asses their fox p3 expression. And tregs should be foxp3 pos by definition. This strategy should be revised or at least discussed.

Response: In the entire manuscript, T_{regs} were always gated on FOXP3+ cells. A majority of those are CD25^{high}, but few cells are and not. Se we have mentioned CD25+, but they were always FOXP3+. For sorting we have additionally used CD127 marker, using a kit. Purifying T_{regs} using this kit allows us to obtain FOXP3+ cells ~ 85% purity. As mentioned in the methods, PD-1^{high} CD25+ cells purified from HIV-infected cultures were 80-88% FOXP3+ and were used as PD-1^{high}CD25+FOXP3+ cells. We have made it clearer in the methods. See line 620.

2. The authors should assess the internalization of IL-10R and modulation of downstream targets, which was published to happen in the intestinal mucosa in SIV infected animals by Pahar, Veazey and Lackner doi: 10.1128/JVI.01757-14

Response: Similar to the above-referenced study here, (Pan2014-SIV mucosa IL10 and IL-10R) our transcriptomics show that IL-10 is increased in oral mucosa HIV+ individuals (Fig. 1B, 2C, 8C). The study also showed an increase in the surface and intracellular IL-10R in jejunum lamina propria, but not jejunum intraepithelial T cells during late acute SIV infection. Also, IL-10R signaling in T_{regs} has previously been shown to be required for suppression of effector CD4+ T cells (Chaudhry et., al,2011, Interleukin-10 Signaling in Regulatory T Cells Is Required for Suppression of Th17 Cell-Mediated Inflammation, Immunity 34, 566–578, DOI 10.1016/j.immuni.2011.03.018). Therefore, as the reviewer suggested, we examined the expression of the IL-10R receptor but did not find changes in IL-10R expression with and without anti-retroviral inhibitor in HIV-infected CD4+ T cells (Fig.R10). We have included these data in Fig.S6E.

Fig.R10. IL-10 receptor was unaltered in CD4⁺ cells in HIV-infected cultures . Purified CD4⁺ cells were activated with TCR stimulation and allowed to expand in the presence of TGF-β1 (10 ng/ml) and IL-2 (100 U/ml). They were infected with HIV on day 2 after TCR stimulation. Flow cytometry was performed to determine IL-10 receptor expression in CD4⁺ cells on 6 days post-infection.

3. Is it possible that conversion of CD4 to pTreg phenotype could add to the change in frequencies? AKT activation is linked to pTreg differentiation.

Response: We verified this possibility by infecting CD4⁺CD25^{neg} CD45RO^{neg} cells and infecting them with HIV under iT_{reg} inducing conditions. While a few cells expressed FOXP3 in the presence and absence of HIV, HIV infection did not upregulate IFN-γ and PD-1 in a significant manner, showing that naïve CD4⁺ T cell conversion may not be a major source of PD-1^{hi} IFN-γ⁺ AREG^{high} FOXP3⁺ cells during HIV infection (Fig.R11).

Fig.R11. Naïve CD4⁺ cells induced in to iT_{reg} cells do not upregulate PD-1 and IFN-γ in HIV-infected cultures. Purified CD4⁺CD25^{neg} CD45RO^{neg} cells were activated with TCR stimulation, TGF-β1 (20 ng/ml) and IL-2 (100 U/ml). They were infected with HIV on day 2 after TCR stimulation. Flow cytometry was performed to determine PD-1, IFN-γ in CD4⁺FOXP3⁺ cells on 6 days post-infection.

4. They should discuss the role of AKT/FOXO3 and IFN-γ in Treg suppressive pathway and potentially look towards multiple sclerosis where they've shown that constitutively active AKT leads to increased IFN production instead of decreasing it and blocking the pathway suppresses IFN and recovers suppressive abilities of Tregs.

doi.org/10.15252/embr.201541905 This can also be an experiment. They've already done a suppression assay, they should do it again, but blockade the pathway or IFN- γ secretion and see if it fixes the suppressive functionality.

Response: This is an important point. As the reviewer suggested, we isolated PD-1^{high}CD25⁺ T_{regs} from HIV-infected cultures and co-cultured with responding (T_{resp}) CD4⁺ T cells (as in Fig.7) but blocking IFN- γ using an anti-IFN- γ antibody. As controls, we also had T_{resp} alone with and without IFN- γ blocking (Fig.R12). However, in the presence of IFN- γ blocking, their proliferation was not reduced when compared to those in PD-1^{high} T_{reg} co-cultures, showing that blocking IFN- γ alone did not restore their suppressive capacity in vitro. Interestingly, we found that blocking IFN- γ moderately blocked the proliferation of T_{resp} even without T_{regs}. This piece of data supports a previous study that showed that IFN- γ is important for TFH accumulation (Lee et., al, 2012, Interferon- γ excess leads to pathogenic accumulation of follicular helper T Cells and germinal centers, Immunity 37, 880-892).

We have also discussed the constitutive Akt activation in modulating T_{regs} in the context of multiple sclerosis in the revised version of the manuscript. See line 475.

Fig.R12

Fig.R12. Loss of suppressive activity in PD-1⁺ FOXP3⁺ IFN- γ ⁺ cells is not restored by blocking IFN-g. Purified CD4⁺ T cells and naïve T_{regs} were stimulated and infected as in methods. PD-1^{hi}CD25⁺ cells were purified from HIV-infected CD4 cultures and were used in co-cultures with cell-trace FarRed labelled responder T cells (T_{resp}) at ratio 1:1 in the presence of anti-IFN- γ antibody, or mouse IgG1 isotype control (10 μ g/ml). As controls, T_{resp} cells were cultured alone or co-cultured with purified T_{regs} that were previously cultured in the same manner. T_{resp} proliferation was determined by cell-trace dye dilution.

5. Further, they should discuss the inherent survival advantage that Tregs have over CD4 during HIV infection due to their difference in apoptotic gene expression

Response: We have discussed this with respect to their high expression of BCL-2 in PD-1 high cells and the previous literature that have shown inherent differences of T_{regs} to resist apoptosis. (See line 496).

6. They should proof the paper again to reduce the incidence of punctuation errors (primarily spacing).

Response: *We have proof-read the revised version to fix the punctuation errors.*

REVIEWERS' COMMENTS

Reviewer #2 (Remarks to the Author):

In the revised manuscript, "PD-1 dependent expansion of Amphiregulin+FoxP3+ cells is associated with oral immune dysfunction in HIV patients on therapy" by Bhaskaran et al., the authors performed a range of very nice and convincing experiments.

Critically, the authors isolate "naïve" Tregs and demonstrated that dysfunctional PD-1 & AREG expressing Tregs can be induced from naïve Treg cells in vitro. These are critical data and enhance the credibility of the presented data substantially.

Unfortunately, the authors did not address the aspect that the measurement of Treg suppressive capacity after four days of in vitro culture is entirely unreliable. (the publications the authors refer to as they had used a similar approach also argued that in vitro Tregs mainly suppress immune responses by sucking up IL-2 – an aspect which is certainly correct, but for the physiological relevance in HIV infected persons less of relevance).

Therefore, I would like to request that the authors mention at appropriate places in the text that the suppressive capacity of isolated Tregs might have been altered during in vitro culture.

Reviewer #3 (Remarks to the Author):

The authors have addressed most of the concerns. There is still no direct demonstration of functional impairment in ex vivo cells and the authors did not provide additional insights into the technical issues that prevented this analysis. The authors do provide new data showing difference's in Bcl-2 staining but these do not appear to be in the manuscript (F8). Finally, there remain a number of editing issues, some of which are described below.

Minor comments

Line 37 Concurrently seems like wrong word.

Line 179-181 Sentence needs editing.

Line 201 Should refer to figure 3F.

Line 265 Remove "infection"

Line 274 Editing needed

FS17 legend Should "some cells were infected" be just "cells were infected" or "cultures were infected"? Same for S18

Reviewer #4 (Remarks to the Author):

The authors addressed my previous comments to my satisfaction. The current version of the manuscript is significantly improved.

Cristian Apetrei

PONT-TO-POINT- REPSONSE TO REVIEWERS' COMMENTS (REVISION-II-MAY, 11 2021)

Reviewer #2 (Remarks to the Author):

In the revised manuscript, “PD-1 dependent expansion of Amphiregulin+FoxP3+ cells is associated with oral immune dysfunction in HIV patients on therapy” by Bhaskaran et al., the authors performed a range of very nice and convincing experiments.

Critically, the authors isolate “naïve” Tregs and demonstrated that dysfunctional PD-1 & AREG expressing Tregs can be induced from naïve Treg cells in vitro. These are critical data and enhance the credibility of the presented data substantially.

Unfortunately, the authors did not address the aspect that the measurement of Treg suppressive capacity after four days of in vitro culture is entirely unreliable. (the publications the authors refer to as they had used a similar approach also argued that in vitro Tregs mainly suppress immune responses by sucking up IL-2 – an aspect which is certainly correct, but for the physiological relevance in HIV infected persons less of relevance).

Therefore, I would like to request that the authors mention at appropriate places in the text that the suppressive capacity of isolated Tregs might have been altered during in vitro culture.

Author's response:

Thank you for the comments. As mentioned by the reviewer, we have included sentences about the *in-vitro* cultured T_{regs} and their potential irrelevance to physiological scenario on:

Pg16; line 346

Pg 16; line 364

Pg 22; line 495

Reviewer #3 (Remarks to the Author):

The authors have addressed most of the concerns. There is still no direct demonstration of functional impairment in ex vivo cells and the authors did not provide additional insights into the technical issues that prevented this analysis. The authors do provide new data showing difference's in Bcl-2 staining but these do not appear to be in the manuscript (F8). Finally, there remain a number of editing issues, some of which are described below.

Author's response:

Thank you for the positive comments.

Gingival biopsies are too small to obtain enough T_{regs} for ex vivo suppressive assays, and these assays require at least 1-2 million purified T_{regs}. We have noted this limitation on pg 16, line 366.

We have included sentences about the *in-vitro* cultured T_{regs} and their potential irrelevance to physiological scenario on:

Pg16; line 346

Minor comments

Line 37 Concurrently seems like wrong word.

Author's response:

We have changed this word to "Concordantly"

Line 179-181 Sentence needs editing.

Author's response:

The sentence is edited.

Line 201 Should refer to figure 3F.

Author's response:

We have changed this to include 3F.

Line 265 Remove "infection"

Author's response:

We have removed the word.

Line 274 Editing needed

Author's response:

The sentence is edited.

FS17 legend Should "some cells were infected" be just "cells were infected" or "cultures were infected"? Same for S18

Author's response:

The sentence was changed to "cultures were infected" in all instances.

Reviewer #4 (Remarks to the Author):

The authors addressed my previous comments to my satisfaction. The current version of the manuscript is significantly improved.

Author's response:

We are grateful to the reviewer for their positive comments. Thank you.